# Specific lid-base contacts in the 26s proteasome control the conformational switching required for substrate degradation

Eric R Greene[1,2†], Ellen A Goodall[1,2†], Andres H de la Peña[3‡], Mary E Matyskiela[1,2‡], Gabriel C Lander[3], Andreas Martin[1,2,4*]

[1]Department of Molecular and Cell Biology, University of California, Berkeley, Berkeley, United States; [2]California Institute for Quantitative Biosciences, University of California, Berkeley, Berkeley, United States; [3]Department of Integrative Structural and Computational Biology, The Scripps Research Institute, La Jolla, United States; [4]Howard Hughes Medical Institute, University of California, Berkeley, Berkeley, United States

**Abstract** The 26S proteasome is essential for proteostasis and the regulation of vital processes through ATP-dependent degradation of ubiquitinated substrates. To accomplish the multi-step degradation process, the proteasome's regulatory particle, consisting of lid and base subcomplexes, undergoes major conformational changes whose origin is unknown. Investigating the *Saccharomyces cerevisiae* proteasome, we found that peripheral interactions between the lid subunit Rpn5 and the base AAA+ ATPase ring are important for stabilizing the substrate-engagement-competent state and coordinating the conformational switch to processing states upon substrate engagement. Disrupting these interactions perturbs the conformational equilibrium and interferes with degradation initiation, while later processing steps remain unaffected. Similar defects in early degradation steps are observed when eliminating hydrolysis in the ATPase subunit Rpt6, whose nucleotide state seems to control proteasome conformational transitions. These results provide important insight into interaction networks that coordinate conformational changes with various stages of degradation, and how modulators of conformational equilibria may influence substrate turnover.

*For correspondence:
a.martin@berkeley.edu

[†]These authors contributed equally to this work

**Present address:** [‡] Celgene Corporation, San Diego, United States

## Introduction

The 26S proteasome is the principal ATP-dependent protease in eukaryotic cells and responsible for the majority of targeted protein turnover, both through the degradation of short-lived regulatory proteins and the clearance of damaged or misfolded polypeptides for protein-quality control (*Hershko and Ciechanover, 1998*). Ubiquitin ligases mark obsolete proteins with poly-ubiquitin chains and thereby target them to ubiquitin receptors on the 26S proteasome, which represents the last component of the ubiquitin-proteasome system and mechanically unfolds, deubiquitinates, and translocates protein substrates into an internal chamber for proteolytic cleavage (*Bard et al., 2018a*). To accomplish these various tasks of substrate processing, the 26S proteasome undergoes significant conformational rearrangements whose origin and control still remain largely elusive.

At the center of the 26S proteasome is a barrel-shaped core peptidase with sequestered proteolytic active sites (*Groll et al., 1997*). This core is capped on one or both ends by a regulatory particle that consists of two subcomplexes, referred to as the "lid" and the "base", and is responsible for the recognition, unfolding, and transfer of protein substrates into the core (*Bard et al., 2018a*;

*Glickman et al., 1998a*). The base contains 10 subunits, including three ubiquitin receptors, Rpn1, Rpn10, and Rpn13, the large scaffolding subunit Rpn2, and six distinct ATPases that form a ring-shaped, heterohexameric AAA+ (ATPase Associated with various cellular Activities) motor in the order Rpt1-Rpt2-Rpt6-Rpt3-Rpt4-Rpt5 (*Tomko et al., 2010*). These ATPases dock on top of the core peptidase to open its gate for substrate transfer (*Smith et al., 2007*). As in other protein unfoldases of the AAA+ family, the six Rpt subunits in the proteasome base use loops with conserved aromatic residues projecting into the central pore of the hexamer to interact with the substrate polypeptide, mechanically pull on it, and drive its translocation into the 20S core in an ATP hydrolysis-dependent manner. These loops lie deep in the pore, such that appropriate substrates require not only a ubiquitin modification for binding to a proteasomal receptor, but also a flexible initiation region of 20–25 residues to reach and engage with this AAA+ translocation machinery (*Prakash et al., 2004*; *Bard et al., 2019*).

The nine-subunit lid binds to one side of the base and thus further expands the regulatory particle's asymmetry contributed by the heterohexameric ATPase ring. The lid includes the $Zn^{2+}$-dependent deubiquitinase (DUB) Rpn11 (*Glickman et al., 1998b*; *Yao and Cohen, 2002*; *Verma et al., 2002*) in a hetero-dimeric complex with another MPN-domain containing subunit, Rpn8 (*Worden et al., 2014*), as well as six scaffolding subunits, Rpn3, 5, 6, 7, 9, and 12. In addition to lid contacts with Rpn2 and the N-terminal regions of Rpt3 and Rpt6, the subunits Rpn5, Rpn6, and Rpn7 use their N-terminal TPR domains to specifically interact with the ATPase domains of Rpt4, Rpt3, and Rpt6, respectively (*Lander et al., 2012*; *Lasker et al., 2012*). Rpn5 and Rpn6 also contact the core peptidase and thus appear to form an external scaffold bridging the lid, base, and core subcomplexes within the proteasome holoenzyme (*Lander et al., 2012*; *Matyskiela et al., 2013*).

Previous cryo-electron microscopy studies identified multiple proteasome conformations with distinct relative orientations and contacts of base, lid, and core that are structurally conserved between yeast and human proteasomes (*Matyskiela et al., 2013*; *Śledź et al., 2013*; *Unverdorben et al., 2014*; *Wehmer et al., 2017*; *Eisele et al., 2018*; *Ding et al., 2017*; *de la Peña et al., 2018*; *Dong et al., 2019*). In the absence of substrate, the proteasome exists in two conformations, s1 and s2, in which Rpt1-Rpt6 form a spiral staircase arrangement with Rpt3 in the top position. In the s1 state, the ATPase ring is not coaxially aligned with the core peptidase, and Rpn11 is offset from the central processing channel, allowing substrate access to the pore entrance. In contrast, the s2 state is characterized by a rotated lid position relative to the base and a coaxial alignment of Rpn11, the ATPase ring, and the core peptidase (*Unverdorben et al., 2014*). Substrate engagement induces conformations that are overall very similar to s2, with a continuous central channel for efficient substrate translocation and a centrally aligned Rpn11 that leaves only a small gap to the subjacent Rpts for substrate to be pulled through, facilitating co-translocational deubiquitination (*Matyskiela et al., 2013*; *de la Peña et al., 2018*; *Dong et al., 2019*). These substrate-processing states, named s3-s6 (*Matyskiela et al., 2013*; *Unverdorben et al., 2014*; *Wehmer et al., 2017*; *Eisele et al., 2018*; *de la Peña et al., 2018*; *Dong et al., 2019*), show AAA+ motor conformations in which various Rpts adopt the individual vertical position in the spiral staircase, depending on the progression of the ATP-hydrolysis cycle in the hexamer. The s5 state thereby resembles s2, with the exception of the core gate that is open in s5 and closed in s2 (*Eisele et al., 2018*). Similar suites of substrate-engaged-like conformations can also be induced by incubating the proteasome with non-hydrolyzable ATP analogs (*Śledź et al., 2013*; *Wehmer et al., 2017*; *Ding et al., 2017*) or introducing Walker-B mutations (*Eisele et al., 2018*), both of which trap Rpts in the ATP-bound state and stabilize their interface to neighboring ATPase subunits in the hexamer. Our recent studies on the coordination of proteasomal degradation steps suggested that substrate engagement depends on the s1 state, in which the entrance to the central pore is accessible and the initiation region of a ubiquitin-receptor-bound substrate would be able to enter the AAA+ motor (*Bard et al., 2019*). Premature switching to substrate-processing states seemed to prevent this substrate engagement, potentially due to Rpn11 obstructing the central pore. Yet alternative models could not be completely ruled out, because the substrate-processing states in those studies were induced by the addition of ATPγS (*Bard et al., 2019*), which abolishes translocation and may also interfere with substrate engagement.

Mutational studies, in which nucleotide binding or hydrolysis of single ATPase subunits were disrupted by substitutions in the Walker-A or Walker-B motifs demonstrated the functional asymmetry of the proteasomal AAA+ motor, as the same mutations in different Rpts caused varied effects on cell viability and the degradation of ubiquitinated substrates (*Eisele et al., 2018*; *Wendler et al.,*

*2012*; *Rubin et al., 1998*; *Beckwith et al., 2013*). However, it remains unclear to what extent these differences in proteasomal activity originate from individual Rpt subunits playing unequal roles in mechanical substrate processing, or from these mutations differentially affecting the overall conformational switching of the proteasome.

Here, we investigate how interactions between the lid and base subcomplexes influence the conformational transitions and thus substrate processing by the *Saccharomyces cerevisiae* 26S proteasome. Previous structural studies showed that the contacts between Rpn5's TPR domain and the small AAA+ subdomain of Rpt3 are broken during the regulatory particle's transition from the substrate-free s1 state to any other state (*Matyskiela et al., 2013*; *Unverdorben et al., 2014*; *Wehmer et al., 2017*; *Eisele et al., 2018*; *Ding et al., 2017*; *de la Peña et al., 2018*; *Dong et al., 2019*). Through mutations of the involved residues in Rpn5, we found that loss of these interactions perturbs the conformational landscape and allows the proteasome to more strongly populate substrate-engaged-like conformations even in the absence of substrate. Walker-B mutations that prevent ATP hydrolysis in individual subunits of the AAA+ motor similarly disrupt the conformational equilibrium. In both cases, perturbing the coordination between substrate-processing steps and conformational transitions of the proteasome's regulatory particle leads to decreased degradation rates, primarily by affecting the initiation of processing and shifting the rate-limiting step from substrate unfolding to engagement. Our data thus reveal how the proteasome uses the peripheral interactions with the lid subunits to orchestrate the conformational transitions required for the various stages of ubiquitin-dependent substrate degradation.

## Results

### The lid is required for proteasome function independent of deubiquitination

Structural rearrangements, specifically the rotation of the lid relative to the base observed in response to substrate processing or binding of ATP analogs to the AAA+ motor, suggest that the lid may be directly involved in determining the proteasome conformational states (*Matyskiela et al., 2013*; *Unverdorben et al., 2014*; *Wehmer et al., 2017*; *Eisele et al., 2018*; *Ding et al., 2017*; *de la Peña et al., 2018*; *Dong et al., 2019*). However, the lid's structural importance for degradation cannot simply be tested by eliminating this subcomplex from the holoenzyme, as it contains the essential DUB Rpn11 and is indispensable for efficient ubiquitin-dependent substrate turnover (*Verma et al., 2002*). We therefore used our previously established ubiquitin-independent substrate-delivery system, in which the bacterial SspB adaptor fused to Rpt2 allows the recruitment of model substrates containing the ssrA recognition motif (*Bashore et al., 2015*). Degradation was monitored through the decrease in anisotropy of a titin-I27$^{V15P}$ model substrate that contained a destabilizing V15P mutation, fluorescein conjugated to the N-terminus, and a C-terminal 35 amino-acid initiation region derived from cyclin B that also included the ssrA recognition motif (FAM-titin-I27$^{V15P}$). Even though we eliminated the dependence on Rpn11-mediated deubiquitination, presence of the lid was still required for efficient ATP-dependent degradation (*Figure 1A*; *Figure 1—figure supplement 1A and B*). In contrast to other compartmental proteases, the proteasomal AAA+ motor and the 20S core peptidase together are not sufficient to catalyze ATP-dependent protein unfolding and degradation.

Interactions between the lid and the AAA+ motor have been found to change in the various proteasome conformations, and these changes of contact points thus represent a possible mechanism by which the lid could act allosterically with the base to influence the regulatory particle's conformational switching during substrate processing. Of particular interest was the contact between Rpn5's TPR domain and Rpt3's small AAA+ subdomain that is present only in the substrate-free s1 state (*Figure 1B*; *Matyskiela et al., 2013*; *Unverdorben et al., 2014*; *Wehmer et al., 2017*; *Eisele et al., 2018*). Mutating all residues in the Rpt3-contacting loop of Rpn5 (V125 - F131) to alanine (mutant denoted Rpn5-VTENKIF) decreased the rate of both, ubiquitin-dependent (*Figure 1C*, *Figure 1—figure supplement 1B and C*) and ubiquitin-independent degradation (*Figure 1—figure supplement 1A and B*). Importantly, this loss of degradation activity is not primarily caused by defects in proteasome assembly, which was found by native PAGE to be only slightly less efficient for the mutant compared to the wild-type enzyme (*Figure 1—figure supplement 2*). Furthermore, using

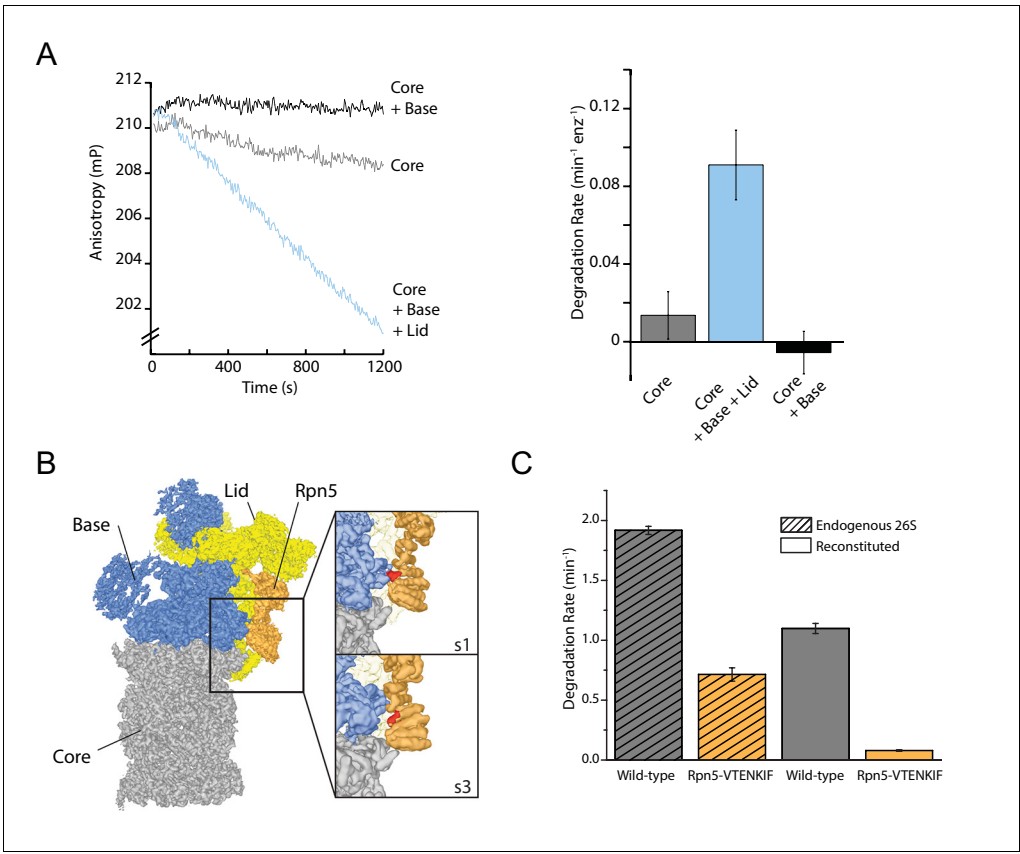

**Figure 1.** The proteasome lid subcomplex is required for proteasome function through direct contacts with the AAA+-motor. (**A**) Ubiquitin-independent degradation of fluorescein-labeled, ssrA-tagged FAM-titin-I27$^{V15P}$ substrate by in-vitro reconstituted 26S proteasomes with recombinantly produced SspB-fused base in the absence and presence of recombinantly produced lid subcomplex was monitored through fluorescence anisotropy under multiple-turnover conditions. Shown on the left are representative traces of changes in anisotropy, and shown on the right are the rates of degradation calculated from these data (N = 3, technical replicates, error bars plotted are SEM). (**B**) Cryo-EM structure of the 26S proteasome from *S. cerevisiae* (EMDB code: 3534) highlights contacts between the lid (yellow and orange), base (blue), and core (grey). The lid subunit Rpn5 (orange) uses a VTENKIF-sequence-containing loop (red) to interact with the small AAA+ subdomain of the base subunit Rpt3 in the substrate-free s1 conformation, but not in any other conformation, like s3 shown here (EMDB: 4321). (**C**) Rates for the single-turnover degradation of a ubiquitinated, TAMRA-labeled G3P substrate with 54 amino acid tail derived from cyclin-b sequence (TAMRA-G3P) by wild-type and Rpn5-VTENKIF mutant proteasomes that were purified from *S. cerevisiae* (shaded) or in-vitro reconstituted using recombinant lid and base (solid) (N = 3, technical replicates, error bars plotted are SEM).

The online version of this article includes the following source data and figure supplement(s) for figure 1:

**Figure supplement 1.** Presence of the lid subcomplex and Rpn5's contact with AAA+ are necessary for substrate processing.

**Figure supplement 1—source data 1.** Source data for substrate degradation by endogenous and reconstituted wild-type and Rpn5-VTENKIF mutant proteasomes.

**Figure supplement 2.** Analysis of proteasome mutant assemblies by Native-PAGE.

---

the response of the base ATPase activity to lid binding during holoenzyme assembly, we determined similar affinities for wild-type and Rpn5-VTENKIF mutant lid (*Figure 1—figure supplement 1D*). Despite their lower degradation activity, Rpn5-VTENKIF mutant proteasomes show an elevated ATPase rate in the absence of substrate that increases in response to substrate processing, albeit to a lesser extent than for wild type (*Figure 1—figure supplement 1D*). In agreement with recent findings (*Nemec et al., 2019*), proteasomes containing the Rpn5-VTENKIF mutation more strongly retained the Nas6 assembly chaperone during holoenzyme reconstitution (*Figure 1—figure*

*supplement 1E*). However, this presence of Nas6 is not the main cause for the observed decrease in degradation rate, as purified endogenous proteasomes from *S. cerevisiae* carrying the same Rpn5 mutations also exhibit major deficiencies in single-turnover degradation reactions (*Figure 1C*), despite containing only negligible amounts of Nas6 (*Figure 1—figure supplement 1F*). Moreover, we found that substrate processing efficiently evicts most Nas6 from nascent Rpn5-VTENKIF mutant proteasomes (*Figure 1—figure supplement 1E*), indicating that the initial Nas6 retention is not responsible for the steady-state substrate processing defect of the Rpn5-VTENKIF mutant. Overall, the observed decrease in degradation activity appears to principally be caused by intrinsically compromised substrate processing, rather than the observed minor defects in the assembly or composition of proteasome holoenzymes (*Figure 1—figure supplements 1* and *2*).

## Lid-base contacts influence proteasome conformation

We employed negative-stain electron microscopy to assess the conformational states of the proteasome and whether the Rpn5-VTENKIF mutation affects their distribution. Because the 20S core can be singly- or doubly-capped by regulatory particles, proteasome particles were half-masked to treat each regulatory particle independently for data processing (*Figure 2—figure supplements 1* and *2*). Consistent with previous observations, ATP-bound wild-type proteasomes in the absence of

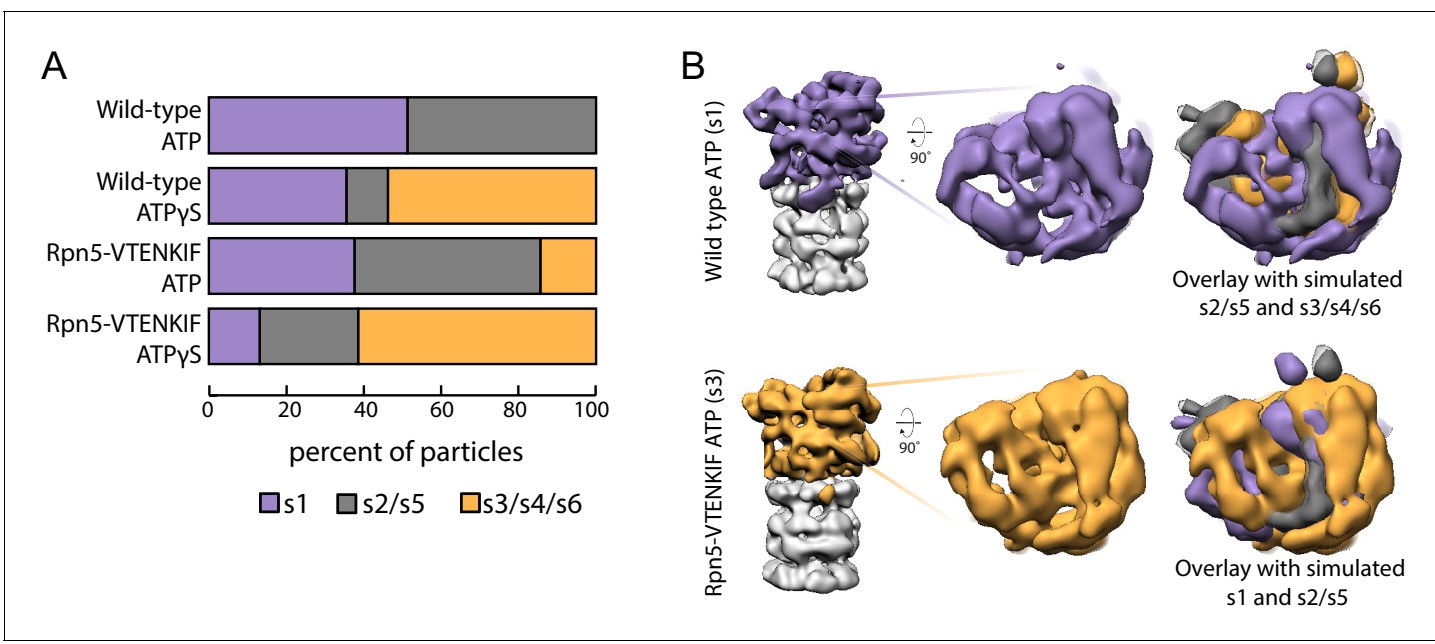

**Figure 2.** Rpn5 interactions with the AAA+ ring are required for maintenance of the s1 state. (**A**) Proportion of each proteasome conformation observed by negative-stain electron microscopy for wild-type or Rpn5-VTENKIF proteasome in the presence of ATP or ATPγS. Designation of substrate-free and engaged-like conformations (s1, s2/s5, s3/s4/s6) was based on best fit to the atomic models provided in *Eisele et al. (2018)* and more details of the classification are provided in *Figure 2—figure supplement 1–6*. (**B**) Representative densities for wild-type proteasome in the s1 conformation (top, purple) and Rpn5-VTENKIF mutant proteasome in the s3/s4/s6 conformation (bottom, orange), overlaid with low-resolution envelopes generated from the atomic models for the given state in *Eisele et al. (2018)* aligned by their core particles (grey). In the overlay s2/s5 is shown in grey, s3/s4/s6 is shown in orange, and s1 is shown in purple.

The online version of this article includes the following source data and figure supplement(s) for figure 2:

**Source data 1.** Source data for EM densities of mutant and wild-type proteasomes in the presence of ATP or ATPγS, and comparison with the s1-s6 conformations from *Eisele et al. (2018)*.

**Figure supplement 1.** EM processing of Rpn5-VTENKIF and wild-type proteasomes.

**Figure supplement 2.** Picking of final 3D classes.

**Figure supplement 3.** Overlays of each class for wild-type proteasomes to simulated 20 Å maps of s1, s2, and s3 states.

**Figure supplement 4.** Low resolution simulated maps from *Eisele et al. (2018)* capture the conformational changes that occur in the regulatory particle.

**Figure supplement 5.** Overlays of each class for Rpn5-VTENKIF-containing proteasomes to simulated 20 Å maps of s1, s2, and s3 states.

**Figure supplement 6.** Overlays of negative-stain classes.

substrate were observed in two conformations, s1 and s2 (*Figure 2A*; *Figure 2—figure supplement 3*; *Bard et al., 2018a*). Despite the limited resolution of negative-stain electron microscopy, these conformations could be distinguished from each other and from the substrate-bound states, in which the lid is even more rotated relative to the base, as obvious from the positioning of the horse-shoe shaped structure formed by the 6 PCI (Proteasome/Cyclosome/eIF3)-domain-containing lid subunits (*Figure 2B*; *Figure 2—figure supplement 4*). In contrast to wild-type proteasomes with nearly equal distribution of s1 and s2 conformations, Rpn5-VTENKIF mutant proteasomes showed only 37% of particles in the s1 state, while also populating substrate-engaged-like states (s3/s4/s6) that are absent from wild-type samples in the presence of ATP (*Figure 2A*, *Figure 2—figure supplements 3*, *5* and *6*). Interestingly, the Rpn5-VTENKIF mutant displayed predominantly s2 or s5 conformations, which similar to s3, s4, and s6 are characterized by Rpn11 obstructing the central pore. The lower population of the s1 state resembles the scenario for Walker-B mutant proteasomes that in recent structural studies were found to have perturbed conformational landscapes as well (*Eisele et al., 2018*). Importantly, however, with bound ATPγS the Rpn5-VTENKIF mutant proteasomes behaved similar to wild-type in shifting to a conformational distribution that is dominated by substrate engaged-like states, which demonstrates their retained ability to conformationally respond when Rpt subunits are trapped in an ATP-bound state.

## The nucleotide states of Rpt6 and Rpt4 affect proteasome conformational switching

Binding of non-hydrolyzable ATP analogs to the proteasomal AAA+ motor triggers similar conformational changes as substrate engagement, suggesting that stabilizing Rpt subunits in an ATP-bound state or coordinating nucleotide binding and hydrolysis in several substrate-interacting Rpt subunits provides a common driving force for conformational transitions. To assess in more detail how perturbations in ATP-hydrolysis affect the conformational states of the proteasome, we placed Walker-B mutations (Glu to Gln) in individual Rpts. Consistent with previous in vitro and in vivo studies that revealed unequal contributions of Rpts to proteasomal degradation activity (*Eisele et al., 2018*; *Beckwith et al., 2013*), we observed differentially reduced rates of substrate turnover for these variants (*Figure 3—figure supplement 1A*), with the strongest defects seen in Rpt subunits that make contacts with TPR domains of lid (Rpt3, Rpt6, and Rpt4, *Figure 3A*). As a readout for their conformational state, we analyzed how Walker-B mutants responded in their ATPase activity to the interaction with ubiquitin-bound Ubp6. Ubp6 is a non-essential, proteasome-interacting DUB that in its ubiquitin-bound form biases the proteasome's conformational equilibrium away from the s1 state and thereby stimulates the ATPase activity similar to substrate processing (*Bashore et al., 2015*; *Peth et al., 2013*; *Aufderheide et al., 2015*). Despite significantly different basal ATPase rates, proteasome variants with a Walker-B mutation in Rpt1, Rpt2, Rpt3, or Rpt5 still maintain some Ubp6-mediated stimulation of ATP hydrolysis (*Figure 3B*, *Figure 3—figure supplement 1B*). However, two mutants with severe degradation defects, Rpt6-EQ and Rpt4-EQ, did not respond to ubiquitin-bound Ubp6 (*Figure 3B*; *Figure 3—figure supplement 1B*), suggesting that a considerable fraction of those proteasomes adopt non-s1 states already in the absence of ubiquitin-bound Ubp6. Additionally, this failure to respond to ubiquitin-bound Ubp6 does not originate from compromised holo-enzyme assembly (*Figure 1—figure supplement 2*; *Figure 3—figure supplement 2*).

Notably, there is a reciprocal crosstalk between the proteasome and Ubp6, in which Ubp6's DUB activity depends on the proteasome conformation, and the highest activity is observed when the catalytic USP domain of Ubp6 interacts with the AAA+ motor in non-s1 states (*Bashore et al., 2015*). All Walker-B mutants except Rpt1-EQ showed increased Ubp6 DUB activity in the presence ATP, but resembled the wild-type enzyme when in ATPγS (*Figure 3C*), confirming their normal conformational response to nucleotide. That Rpt6-EQ and Rpt4-EQ-mutant proteasomes show increased Ubp6 DUB activity in ATP is consistent with their lack of Ubp6-mediated ATPase stimulation and further suggests that trapping Rpt4 or Rpt6 in permanent ATP-bound states populates non-s1 conformations, even in the absence of substrate or ATPγS (*Figure 3C*).

Rpt4-EQ and Rpt6-EQ mutant proteasomes showed strong degradation defects and were consistent in both, Ubp6-mediated ATPase stimulation and Ubp6 DUB activation, indicating a conformational bias away from the engagement-competent s1 state. We therefore tested their core gate-opening activities through fluorogenic peptide hydrolysis in the presence of ATP or ATPγS. Complete opening of the core particle gate requires the docking of five Rpt C-terminal tails: Rpt2, Rpt3,

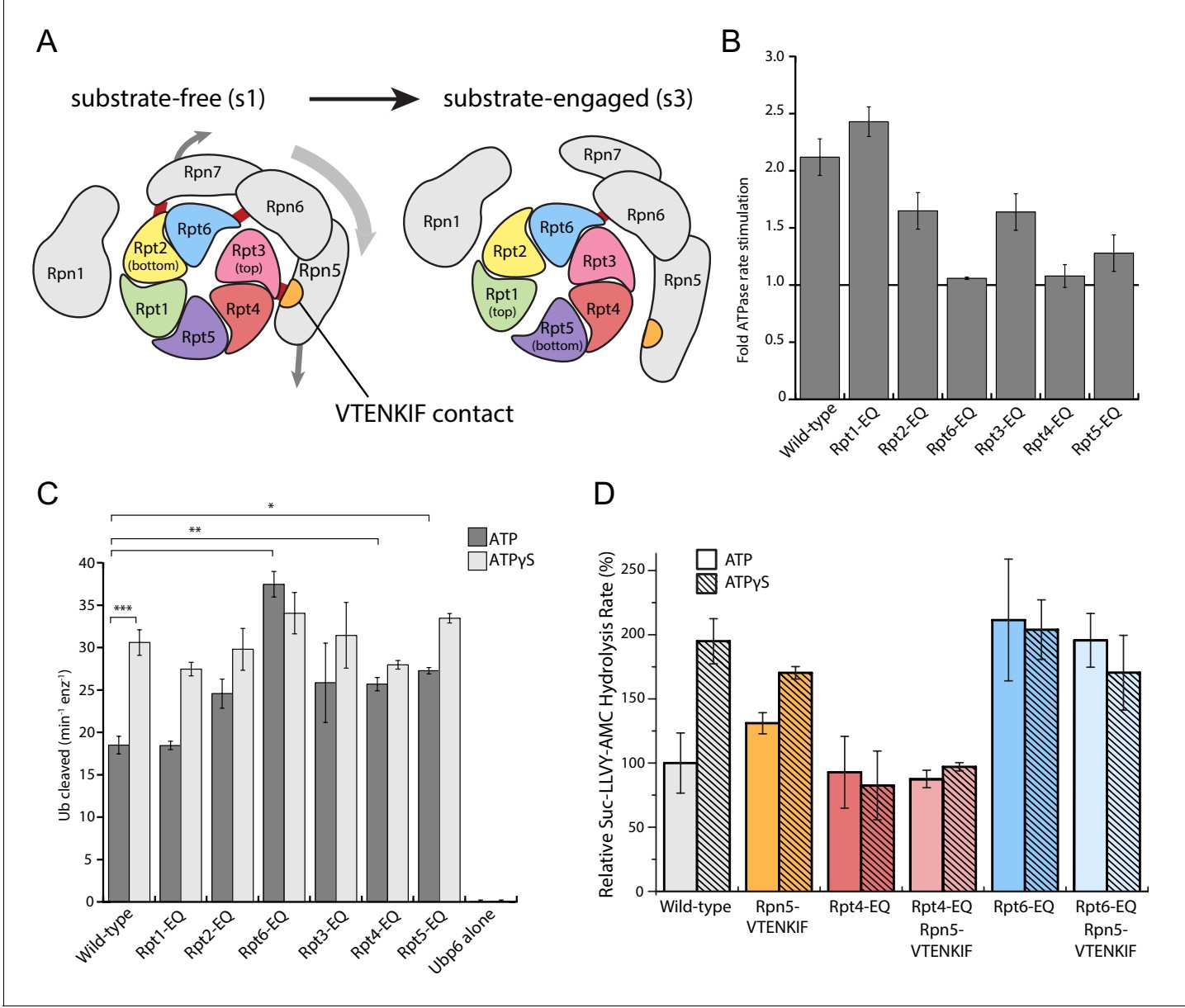

**Figure 3.** Walker-B mutations in Rpt6 and Rpt4 influence proteasome conformational switching and are dominant over Rpn5-VTENKIF mutation. (**A**) Cartoon of the proteasome heterohexameric AAA+-motor and interacting lid subunit as viewed from the top, with the core particle underneath. The base subunit Rpn1 and lid subunits are shown in grey, with red bars indicating the interactions between the VTENKIF region of Rpn5 (orange) and the AAA+ domain of Rpt3 (pink), as well as the Rpn6-Rpt6 and Rpn7-Rpn2 contacts. The ATPase subunits Rpt1, Rpt2, Rpt6, Rpt3, Rpt4, and Rpt5, depicted in rainbow colors, are forming a vertical spiral staircase. In the substrate-free s1 state, Rpt3 is at the top of this staircase, Rpt2 at the bottom, and Rpt6 represents the seam subunit with an open ATPase interface to its neighbor Rpt3. In the substrate-engaged s3 state, Rpt1 is at the top and Rpt5 at the bottom, with an open seam between the two. As the ATPase ring transitions through the various engaged states during ATP hydrolysis, the staircase and the open seam are expected to progress in a counterclockwise manner around the ring (*de la Peña et al., 2018*). (**B**) Proteasome ATPase stimulation by ubiquitin-bound Ubp6, with no stimulation indicated by a solid black line (N ≥ 3, technical replicates, error bars plotted are SEM). (**C**) Ub-AMC cleavage activities of Ubp6 in the context of wild-type or Walker-B mutant proteasomes with ATP or ATPγS (N ≥ 3, technical replicates, error bars plotted are SEM, p values shown for a Student's T-test). (**D**) Core gate-opening measured through cleavage of the fluorogenic Suc-LLVY-AMC substrate. Cleavage rates were determined by linear fitting of the AMC-fluorescence increase, normalized to wild-type proteasome in ATP, and plotted as averages with standard deviations (N ≥ 3, technical replicates).

The online version of this article includes the following source data and figure supplement(s) for figure 3:

**Figure supplement 1.** Degradation and ATPase activities of Walker-B mutant proteasomes.

**Figure supplement 1—source data 1.** Source data for ATP-hydrolysis, Ubiquitin-AMC-cleavage, and core-gate-opening activities of wild-type and Walker-B (EQ) mutant proteasomes.

*Figure 3 continued on next page*

*Figure 3 continued*

**Figure supplement 2.** Core-gate opening activities of reconstituted proteasomes.

Rpt5, Rpt1, and Rpt6, with the latter two only found fully-docked in substrate engaged-like conformations - that is, during degradation or in the presence of non-hydrolyzable ATP analogs (*Eisele et al., 2018*; *de la Peña et al., 2018*; *Dong et al., 2019*; *Zhu et al., 2018*). The Rpt6-EQ mutant showed elevated gate opening in the presence of ATP that resembled the ATPγS-bound wild-type proteasome and did not further increase upon ATPγS addition (*Figure 3D*), which is consistent with this variant being biased towards an engaged-like conformation due to trapping Rpt6 in a permanent ATP-bound state. Consistently, the Rpn5-VTENKIF mutant proteasome, whose conformational distribution appeared partially shifted in our EM analyses, exhibited moderately increased core gate-opening and peptide-hydrolysis activity that still responded to ATPγS. Proteasomes containing the combined Rpn5-VTENKIF and Rpt6-EQ mutations behaved largely similar to the Rpt6-EQ mutant proteasome, indicating that preventing ATP hydrolysis in Rpt6 and thus stabilizing the interface with Rpt3 has a dominant effect on determining the conformational state, at least with respect to core-particle docking and gate-opening.

Although Rpt4-EQ mutant proteasomes also appeared to be biased towards engaged-like, non-s1 conformations based on their crosstalk with Ubp6, their core-gate opening resembled the ATP-bound wild-type holoenzyme and was not responsive to ATPγS binding (*Figure 3D*). These proteasomes displayed decreased assembly under non-equilibrium conditions in native-PAGE analyses, which could explain some, yet not all of the gate-opening defects, as holoenzyme is clearly formed (*Figure 1—figure supplement 2*). Moreover, the gate-opening activity of the Rpt4-EQ mutant was insensitive to increased base concentrations (*Figure 3—figure supplement 2*), arguing against an assembly defect as the main reason for the functional deficiencies and suggesting that a biased conformational landscape of assembled proteasome is largely responsible for the observed effects. The Rpt4-EQ mutation thus seems to induce a partially distorted conformation that interacts with ubiquitin-bound Ubp6 similar to an engaged-state proteasome, but fails to properly dock with core particle for complete gate opening. Like the Rpt6-EQ mutation, the Rpt4-EQ mutation is dominant in determining the conformational state and therefore masks the stimulating gate-opening effects of the Rpn5-VTENKIF mutation in the combined mutant (*Figure 3D*). Compromising the lid-base interface through Rpn5-VTENKIF mutations thus appears to partially shift the conformational equilibrium of the proteasome, while trapping Rpt6 or Rpt4 in ATP-bound states overrules those changes and further shifts the equilibrium towards either a fully engaged-like or a distorted, potentially off-pathway conformation.

## Proteasomes with biased conformational landscapes display various degradation defects

To understand how these conformation-influencing mutations affect substrate degradation, we first performed Michaelis-Menten kinetic analyses using our ubiquitinated FAM-titin-I27$^{V15P}$ model substrate with a C-terminal 35 amino-acid initiation region that contained a single lysine-attached ubiquitin chain next to the titin folded domain. Rpt4-EQ mutant proteasome showed no discernable degradation activity at any substrate concentrations tested (*Figure 4A*), and further measurements under single-turnover conditions revealed only a small change in anisotropy that we could attribute solely to substrate deubiquitination (*Figure 4B*), as no peptide products were detected in an end-point analysis by SDS-PAGE (*Figure 4C*). These results were confirmed using an additional model substrate, ubiquitinated TAMRA-G3P (*Figure 4—figure supplement 1B*), for which the small amounts of produced peptides could be attributed to nonspecific proteolysis of the unstructured region by the core particle, as previously observed (*Bard et al., 2019*; *Myers et al., 2018*; *Wenzel and Baumeister, 1995*). Furthermore, free Rpt4-EQ-containing regulatory particle, a prominent species in the native-PAGE analysis (*Figure 1—figure supplement 2*), harbored little deubiquitination activity compared to wild-type, Rpn5-VTENKIF, and Rpt6-EQ mutant RPs (*Figure 4—figure supplement 1A–D*). For all wild-type and mutant proteasomes tested, the addition of excess regulatory particle did not change the rate of substrate processing, that is degradation or deubiquitination that would lead to changes in anisotropy (*Figure 4—figure supplement 1A,E*). Interestingly, the

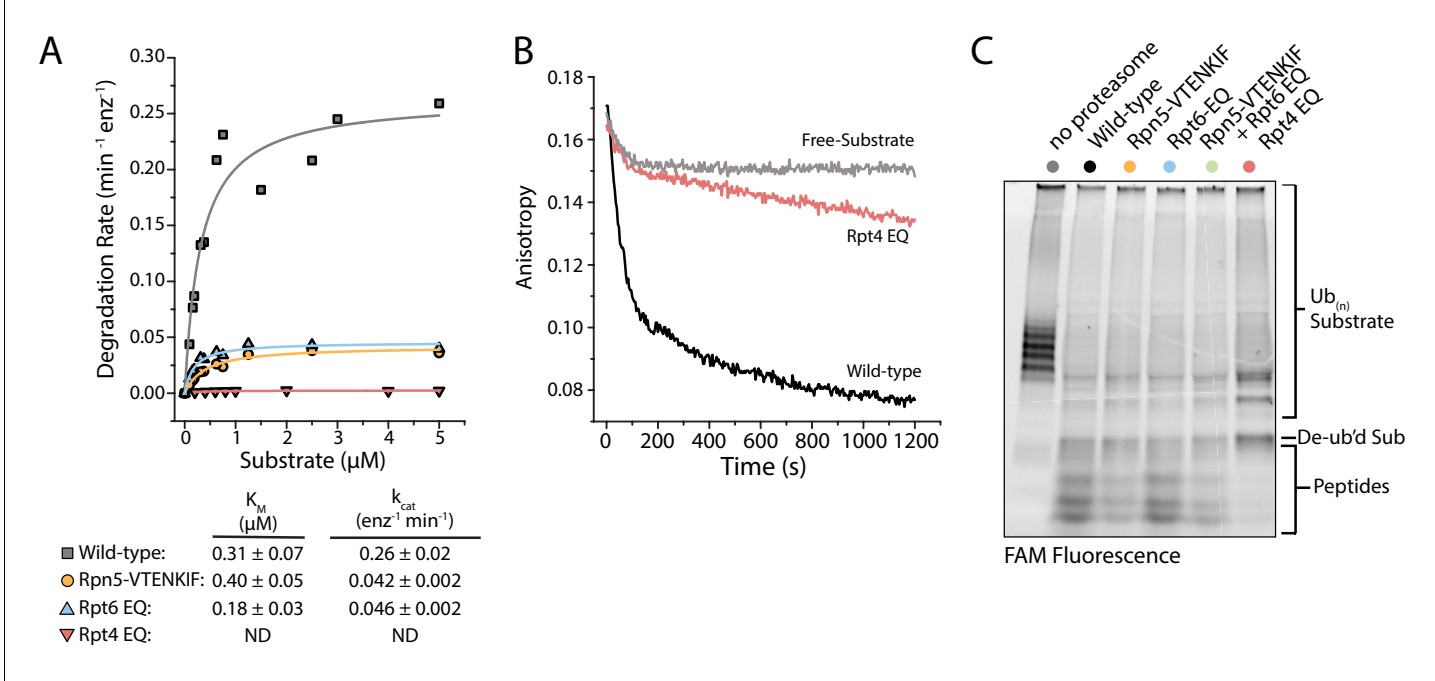

**Figure 4.** Proteasomes with impaired conformational switching display various degradation defects. (**A**) Michaelis-Menten analysis based on initial rates for ubiquitin-dependent degradation of FAM-titin-I27$^{V15P}$ under multiple-turnover conditions. $K_m$ and $k_{cat}$ values are shown below with errors representing SEM from the fit. Rpt4-EQ had too low activity to be fit. (**B**) Representative anisotropy traces for the single-turnover degradation of ubiquitinated FAM-titin-I27$^{V15P}$ by wild-type and Rpt4-EQ mutant proteasomes (**C**) SDS-PAGE analysis of end-point samples from single-turnover degradation reactions, visualizing fluorescence of the FAM-titin-I27$^{V15P}$ model substrate in its ubiquitinated, de-ubiquitinated, and degraded form. The online version of this article includes the following source data and figure supplement(s) for figure 4:

**Figure supplement 1.** Analysis of substrate processing by free regulatory particle during proteasome degradation.

**Figure supplement 1—source data 1.** Source data for Michaelis-Menten analyses of substrate degradation by wild-type, Rpn5-VTENKIF, Rpt4-EQ, and Rpt6-EQ mutant proteasomes, and source data for substrate processing by the corresponding regulatory particles.

deubiquitination activity of the Rpt4-EQ containing regulatory particle increases in a core-dependent manner (*Figure 4—figure supplement 1A*), suggesting that docking to the core particle positions Rpn11 in a more active conformation than in the free mutant RP (*Dambacher et al., 2016*). However, the Rpt4-EQ mutant proteasome lacks substrate engagement or translocation activities, and thus appears degradation-incompetent.

The Rpt6-EQ and Rpn5-VTENKIF mutations decreased the $k_{cat}$ for substrate degradation about 6-fold, with only minimal effects on $K_m$ (*Figure 4A*). This behavior is expected, if these mutations primarily shift the conformational equilibrium and thereby reduce the fraction of engagement-competent s1-state proteasomes. We previously identified tail engagement to be a major determinant of $K_m$ (*Bard et al., 2019*). Based on the lack of major $K_m$ changes, we can thus conclude that the Rpt6-EQ and Rpn5-VTENKIF mutations do not considerably affect substrate engagement of proteasomes in the s1 state (*Bard et al., 2019*). It is assumed that non-s1 states that are not yet substrate-engaged do not significantly contribute to substrate processing, because their coaxially aligned position of Rpn11 right above the entrance to the pore interferes with substrate-tail insertion for degradation and also limits access to the DUB active site for potential translocation-independent deubiquitination. Accordingly, we did not detect deubiquitination and release of unmodified substrate from Rpn5-VTENKIF mutant proteasomes (*Figure 4C*; *Figure 4—figure supplement 1*), which is consistent with our previous findings that non-s1-state proteasomes with bound ATPγS show only low deubiquitination activity towards unengaged protein substrates (*Worden et al., 2017*). It is conceivable that the Rpn5-VTENKIF mutation reduces $k_{cat}$ more strongly than the EM-observed shift in the conformational equilibrium would suggest, if weakening the lid-base interactions increases the dynamics of conformational transitions, and the life time of the engagement-competent s1 state in the mutant proteasomes is shorter than the time constant for substrate-tail insertion (τ = 1.6 s;

*Bard et al., 2019*). The Rpt6-EQ mutant proteasome were previously found to exhibit a similar distribution of s1 and non-s1 states as the Rpn5-VTENKIF mutant here (*Eisele et al., 2018*), and its rates of switching out of and back to the engagement-competent s1 state are expected to be determined by ATP binding and release of the hydrolysis-dead Rpt6 subunit. A 6-fold reduction in $k_{cat}$ compared to wild-type proteasome can thus also be explained by compromised conformational switching and a shorter life time of the s1 state in the presence of the Rpt6-EQ mutation.

## Disrupting the proteasome conformational equilibrium affects degradation initiation

We recently discovered that the engagement of a substrate's unstructured initiation region by the AAA+ motor triggers the major conformational change away from the s1 state, during which the contacts between the base and the VTENKIF-containing loop in Rpn5 are broken (*Bard et al., 2019*). We therefore aimed to investigate how the Rpn5-VTENKIF mutation with its effects on the conformational equilibrium influences this critical step of substrate processing. Using our previously established assay to monitor FRET between a fluorescence donor placed near the central channel of the base and an acceptor fluorophore attached to the substrate, we measured the kinetics of inserting the substrate's flexible tail into the pore (*Bard et al., 2019*). Inhibiting deubiquitination by Rpn11 with the $Zn^{2+}$-chelator ortho-phenanthroline (*o*-PA) stalls further translocation in these experiments and leads to the accumulation of stably engaged substrate in a high-FRET state. Our measurements revealed that tail insertion takes about twice as long for the Rpn5-VTENKIF mutant proteasome compared to wild type (*Figure 5A*, *Figure 5—figure supplement 1A*). We assume that this rate represents a convolution of fast tail insertion for engagement-competent s1-state proteasomes and delayed tail insertion for proteasomes that first have to switch back to the s1 state. The Rpt6-EQ mutant proteasome displayed comparable tail-insertion defects (*Figure 5—figure supplement 1B*), indicating that initial substrate engagement is similarly compromised for both variants, likely due to changes in their conformational landscapes. In agreement with previous findings (*Bard et al., 2019*), very minimal, negligible tail insertion was observed with either proteasome variant in ATPγS or in the absence of core and lid (*Figure 5A*; *Figure 5—figure supplement 1A*).

To determine whether the degradation defects of Rpn5-VTENKIF and Rpt6-EQ mutant proteasomes originate primarily from delayed tail insertion when particles reside in the wrong state or from impaired subsequent processing steps as well, we performed degradation-restart experiments after stalling and accumulating engaged proteasomes through reversible *o*-PA-inhibition of substrate deubiquitination by Rpn11 (*Worden et al., 2017*). Upon release from the stall through the addition of excess $Zn^{2+}$, we monitored the depletion of ubiquitinated TAMRA-G3P substrate as well as the accumulation of peptides products by SDS-PAGE, both of which showed single-exponential behavior (*Figure 5—figure supplement 2A*). As expected, wild-type proteasomes displayed degradation kinetics in the restart experiments that resembled those under non-stalled, single-turnover conditions, because the processing steps preceding the stall, that is tail insertion and the conformational switch upon substrate engagement, are not rate limiting for degradation (*Figure 5B*; *Bard et al., 2019*; *Worden et al., 2017*). Importantly, Rpn5-VTENKIF and Rpt6-EQ mutant proteasomes that showed significant degradation defects under non-stalled, yet otherwise identical conditions, fully regained wild-type degradation rates when restarted after the *o*-PA-induced deubiquitination stall (*Figure 5B*). These data indicate that tail insertion and engagement, but not the subsequent deubiquitination, unfolding, and translocation, are compromised by these mutations, likely through perturbations of the conformational equilibrium and reducing the fraction of proteasomes in the substrate-engagement competent s1 state. The early initiation and commitment steps of degradation are thus strongly dependent on the conformational bias and dynamics of the substrate-free proteasome.

Because a major kinetic deficit for the Rpn5-VTENKIF and Rpt6-EQ mutant proteasomes is incurred at substrate-tail insertion and engagement, degradation by these mutants is likely no longer rate-limited by mechanical unfolding and translocation, in contrast to what is observed for the wild-type proteasome (*Bard et al., 2019*). To address this aspect in more detail, we characterized the ubiquitin-dependent degradation of titin substrates with various thermodynamic stabilities. While wild-type proteasomes degraded the strongly destabilized FAM-titin-I27$^{V13P/V15P}$ variant significantly faster than the non-destabilized FAM-titin-I27, Rpn5-VTENKIF and Rpt6-EQ mutant proteasomes both showed only small differences in degradation for these two substrates (*Figure 5C*). These data indicate that unfolding does not represent the rate-determining step for degradation of the

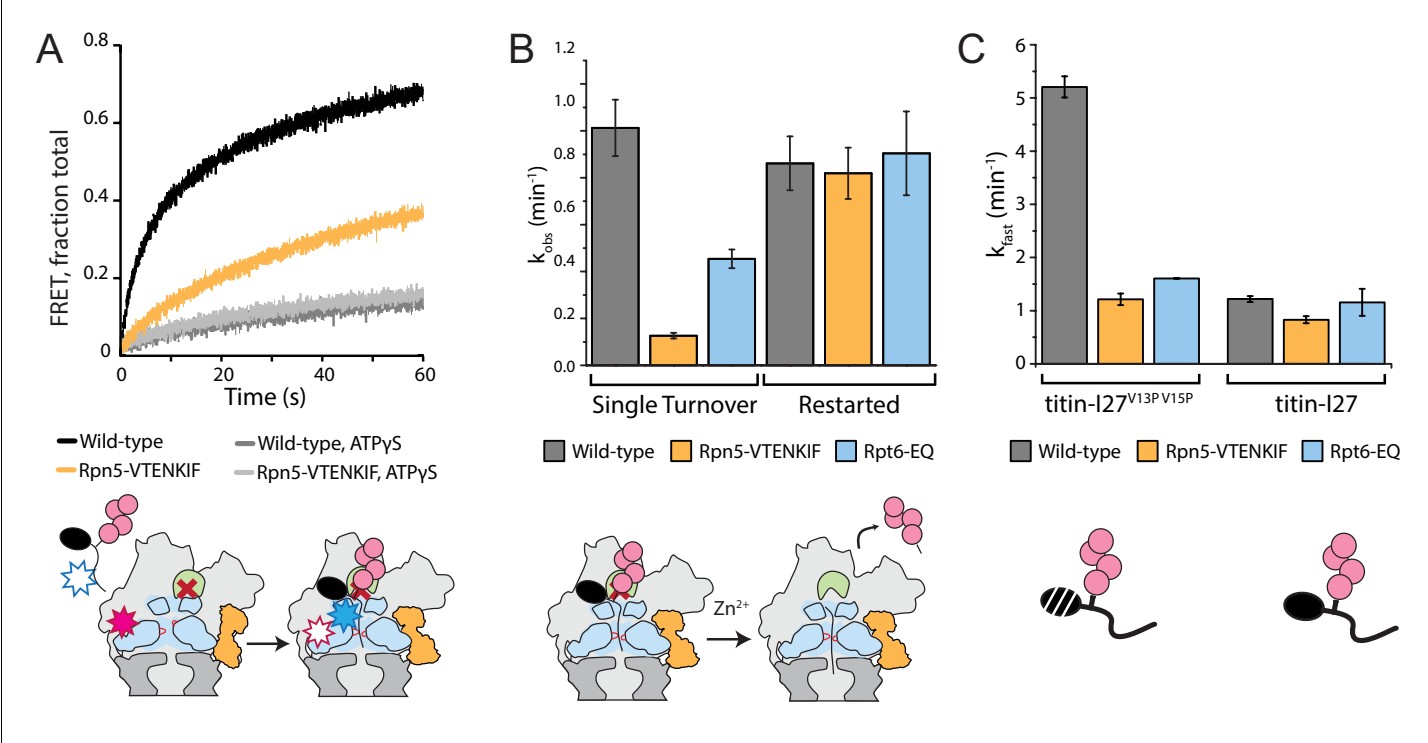

**Figure 5.** Disrupting the conformational equilibrium inhibits substrate tail insertion but not later steps of degradation. (**A**) Representative traces for the increase in acceptor fluorescence/FRET upon insertion of the ubiquitinated FAM-titin-I27$^{V15P}$-Cy5 substrate's flexible initiation into the central pore of wild-type and Rpn5-VTENKIF mutant proteasomes with *o*-PA inhibited Rpn11, in the presence of ATP or ATPγS. The schematic below depicts the experimental setup, where FRET occurs when a substrate's flexible initiation region labeled with an acceptor dye (blue star) enters and then stalls in the central pore of a proteasome containing inhibited Rpn11 (red cross) and a donor dye (red star) near the processing channel. The substrate's ubiquitin modification is represented in pink, the Rpt ring is shown in light blue, the core particle in dark grey, and Rpn5 in orange. (**B**) Rate constants for the single-turnover, ubiquitin-dependent degradation of ubiquitinated G3P model substrate, either without stalling the proteasome (left) or after stalling translocation for 3 min with *o*-PA inhibited Rpn11 and restarting by the addition of $Zn^{2+}$ (right). Rates were determined from single-exponential fits of the appearance of fluorescently tagged peptide products on SDS PAGE gels. Error bars represent SEM for the fit, $N \geq 3$, technical replicates. (**C**) Ubiquitin-dependent degradation rates for wild-type, Rpn5-VTENKIF and Rpt6-EQ mutant proteasomes degrading the destabilized FAM-titin-I27$^{V13P/V15P}$-35mer tail or the non-destabilized FAM-titin-I27-35mer tail substrate under single-turnover conditions. Shown are the rate constants for the dominant fast phase derived from a double-exponential fit of the degradation kinetics (N = 3, technical replicates, error bars represent SD).

The online version of this article includes the following source data and figure supplement(s) for figure 5:

**Figure supplement 1.** Tail insertion can be rate limiting for mutant proteasomes with compromised conformational equilibria.
**Figure supplement 1—source data 1.** Source data for the single-turnover degradation of titin substrates by wild-type, Rpn5-VTENKIF, and Rpt6-EQ mutant proteasomes with or without prior translocation stalling, and source data for substrate-tail insertion into these proteasome variants.
**Figure supplement 2.** Gel based degradation assay analysis.
**Figure supplement 2—source data 1.** Source data for titin-substrate degradation by wild-type, Rpn5-VTENKIF, and Rpt6-EQ mutant proteasomes as analyzed by SDS PAGE.

destabilized titin variants by the mutant proteasomes. Interestingly, the non-destabilized FAM-titin-I27 is degraded by wild-type, Rpn5-VTENKIF, and Rpt6-EQ-mutant proteasomes with comparable rates (*Figure 5C*), suggesting that thermodynamic stability of this substrate is high enough to make mechanical unfolding the common rate-limiting step for all proteasomes. Kinetic SDS-PAGE analysis of the non-destabilized FAM-titin-I27 degradation reaction showed that the decay of ubiquitinated substrate and the appearance of peptide products were anticorrelated, confirming that the observed rates for FAM-titin-I27 processing reflect true degradation and not an aberrant deubiquitination and release process (*Figure 5—figure supplement 2B*). For the Rpn5-VTENKIF and Rpt6-EQ mutant proteasomes this means that the rate-limiting step in degradation changed from initial engagement for more labile substrates to mechanical unfolding for substrates with higher thermodynamic stability. Again, these findings suggest that compromising the conformational equilibrium of

the proteasome primarily affects the early steps of degradation, with no major influence on mechanical unfolding and translocation.

## Discussion

Numerous structural studies of the 26S proteasome have established a suite of conformations that showed various distributions under different conditions (*Matyskiela et al., 2013*; *Unverdorben et al., 2014*; *Wehmer et al., 2017*; *Eisele et al., 2018*; *Ding et al., 2017*; *de la Peña et al., 2018*; *Dong et al., 2019*). Based on those studies, the conformational landscape of the proteasome can be biased by the nucleotide occupancy of the AAA+ motor (*Śledź et al., 2013*; *Unverdorben et al., 2014*; *Wehmer et al., 2017*; *Eisele et al., 2018*; *Ding et al., 2017*; *Zhu et al., 2018*) and the engagement of protein substrates (*Matyskiela et al., 2013*; *de la Peña et al., 2018*; *Dong et al., 2019*), but how the network of contacts within the regulatory particle, and in particular between the lid and base subcomplexes, affects conformational changes and equilibria remained unknown.

Here we report that the lid subcomplex is required for substrate processing independent of the deubiquitination activity contributed by its Rpn11 DUB. The lid subunit Rpn5, whose contact with the base ATPase ring changes dramatically during proteasome conformational changes, plays a critical role in stabilizing the engagement-competent s1 state and coordinating the conformational switch upon substrate engagement by the AAA+ motor. The changes in the conformational landscape caused by the Rpn5-VTENKIF mutation are reminiscent of those incurred by Walker-B mutations in certain Rpt subunits (*Eisele et al., 2018*), for which a significant population of proteasome particles still adopt the s1 conformation, but additional substrate-engaged-like states are accessed as well. Unique to the Rpn5-VTENKIF mutant, however, is the predominance of the s2- or s5-like states, which feature a similar spiral-staircase orientation of the AAA+ motor as the engagement-competent s1 state, but have the ATPase ring and core peptidase coaxially aligned, lack the interaction between the Rpn5-VTENKIF region and Rpt3, and show the lid rotated relative to the base, with Rpn11 obstructing access to the central pore (*Unverdorben et al., 2014*; *Eisele et al., 2018*). ATP-bound wild-type proteasomes also have a fraction of molecules in the s2 conformation (*Bard et al., 2018a*), which is likely adopted through the spontaneous release of lid-base contacts, without rearranging the AAA+ motor staircase. In contrast, the substrate-engaged proteasome conformations are characterized by a multitude of Rpt-staircase arrangements, in addition to having the base coaxially aligned with the core and the lid in a rotated position. This observation suggests that breaking the interactions between Rpn5-VTENKIF and Rpt3, and consequently rotating the lid relative to the base, are likely the first steps in the transition from s1 to substrate-processing states and prerequisites for the staircase re-arrangements of the AAA+ motor. It is conceivable that these peripheral lid-base interactions are disrupted, when several Rpt subunits grab a substrate with their pore loops during engagement and thus become more coordinated in their ATPase cycles, leading to the various spiral-staircase arrangements observed for the substrate-engaged proteasome.

We found that disrupting lid-base interactions and thereby perturbing the conformational landscape of the proteasome leads to significant degradation defects, illustrating the critical importance of the s1 state for substrate-tail insertion and degradation initiation. Previously, we assessed the dependence of substrate engagement on the s1 state by inducing engaged-like conformations through the addition of ATPγS (*Bard et al., 2019*). However, in these studies it could not be completely ruled that, in addition to the conformational bias, shutting down the ATPase motor with the non-hydrolyzable ATP analog also played a role in causing the observed tail-insertion defects. The Rpn5-VTENKIF mutant proteasome characterized here contains a completely unmodified ATPase ring and mutations in Rpn5 that are relevant for subunit interactions exclusively in the s1 state. That this mutant shows strongly compromised degradation initiation therefore provides important new evidence for the s1-state requirement of substrate engagement and the critical role of proteasome conformational changes in coordinating the individual steps of substrate processing. While our EM snapshot of the conformational distribution revealed that Rpn5-VTENKIF mutant proteasomes still retain a considerable fraction of particles in the s1 state, weakening the lid-base interactions may strongly affect the dynamics of conformational switching and shorten the time proteasomes spend in the s1 state. Recent work established a kinetic-gateway model for substrate entry into the proteasome, in which only sufficiently long and complex tails on a substrate are able

to enter the central pore of s1-state proteasomes and trigger the conformational switch to substrate-processing states for degradation (*Bard et al., 2019*). Shortening the life time of the s1 conformer would therefore disrupt this substrate-selection mechanism, interfere with degradation initiation, and lead to major degradation defects, as we observed here. This model does not only apply to the disruption of lid-base contacts, but also to the stabilization of substrate-processing states, for instance by trapping Rpt subunits with bound ATP or ATPγS, which may thus explain the differential degradation defects previously reported for the various Walker-B mutants (*Eisele et al., 2018*; *Beckwith et al., 2013*). In both the s1 and s2 states of substrate-free proteasomes, the ATP-binding pocket of Rpt6 is open and ADP-bound, and the Rpt6-Rpt3 interface acts as the seam in the spiral-staircase arrangement of ATPase subunits, with Rpt3 at the top and Rpt2 at the bottom (*Figure 3A*). In the substrate-processing conformations, however, the Rpt6 pocket is ATP-bound and closed, and may only transiently open up for nucleotide exchange during processive ATP hydrolysis and substrate translocation, similar to all other ATPase subunits in the hexamer (*Figure 3A*). Eliminating ATP hydrolysis in Rpt6 therefore biases the proteasome away from the s1 and s2/s5 states, towards substrate-engaged like conformations (*Eisele et al., 2018*), and is expected to inhibit the progression of the hexamer's sequential ATPases cycle at a stage when the neighboring Rpt3 subunit is in the bottom position of the spiral staircase (*Figure 6*). That trapping Rpt4 with bound ATP also shifts the conformational equilibrium away from the engagement-competent s1 state is somewhat surprising, given that Rpt4's ATPase pocket at the interface with Rpt5 is already closed and ATP-bound in the substrate-free s1 state. Despite their correct assembly into 26S holoenzymes with robust peptidase, ATPase, and deubiquitination activities, Rpt4-EQ mutant proteasomes are degradation-incompetent, which explains the previously described lethality of this mutant in yeast

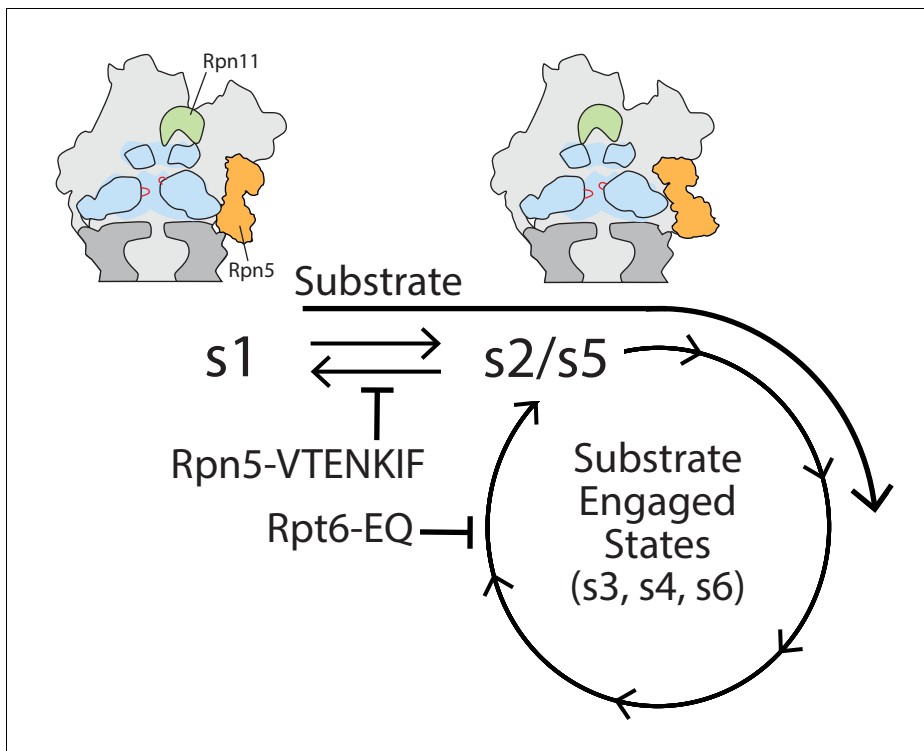

**Figure 6.** Model for coupling between proteasome conformations and substrate degradation. In the s1 conformation, Rpn11 is offset from the central pore, which is therefore accessible for substrate entry and engagement by the AAA+ motor. In the s2/s5 and substrate-engaged states of the proteasome, Rpn11 is coaxially aligned with the continuous processing channel and obstructs the entrance to the AAA+ motor, inhibiting access for substrates that are not yet engaged. Insertion of a substrate's flexible initiation region in the s1 state induces the transition to substrate-engaged conformations. Rpn5-VTENKIF and Rpt6-EQ mutations bias the conformational landscape away from the s1 state, either by destabilizing s1 or trapping substrate-engaged-like states through ATPase inhibition, and therefore interfere with substrate engagement.

(*Eisele et al., 2018*) and highlights the importance of the Rpt4 ATPase pocket for proteasome function. It remains unclear whether the degradation defects of the Rpt4-EQ mutant primarily originate from a perturbed conformational equilibrium that may be more completely or irreversibly shifted to engagement-incompetent non-s1 states, or whether other degradation steps besides initiation are compromised as well.

For the Rpt6-EQ mutant we observed full rescue of degradation defects when proteasomes were restarted after stalling them prior to substrate deubiquitination, indicating that only substrate-tail insertion and engagement by the AAA+ motor were affected by the mutation. The Rpt6-EQ mutant proteasome was previously shown by cryo-EM to adopt substrate-engaged like conformations in the absence of substrate (*Eisele et al., 2018*), which is further supported by their elevated, ATPγS-insensitive core gate-opening activity, their stimulated ATPases rates that are non-responsive to ubiquitin-bound Ubp6, and their increased Ubp6-DUB activity. In agreement with the idea that conformational switching is primarily driven by the nucleotide states and hydrolysis in the motor, the core gate-opening effects contributed by Rpt4-EQ and Rpt6-EQ mutations were unaffected by the addition of the Rpn5-VTENKIF mutation in lid. Conversely, the Rpn5-VTENKIF mutation at the lid-base interface was dominating the degradation defects when combined with Rpt6-EQ, which can be explained if these two mutants have intrinsically different dynamics of conformational switching.

In summary, the detailed characterization of Rpn5-VTENKIF and Rpt6-EQ mutant proteasomes provides insight into the network of interactions within the regulatory particle that govern the crucial conformational switch during degradation. Both mutants highlight the critical importance of the s1 state being populated long enough for substrate-tail insertion and engagement, before the conformational switch to substrate-processing states enables processive threading, mechanical unfolding, co-translocational deubiquitination, and substrate transfer into the 20S core for proteolytic cleavage. While these mutations give rise to similar degradation defects, they are located in distant regions of the proteasome and affect the conformational switching in distinct ways. The Rpn5-VTENKIF mutation disrupts critical lid-base interactions, destabilizes the s1 state, and causes a spontaneous re-equilibration of proteasome conformations. In contrast, the Rpt6-EQ mutation stabilizes substrate-processing states and inhibits the sequential progression of the ATPase hydrolysis cycle in the hexamer, thereby pulling the conformational equilibrium away from the s1 state. These alternative ways of shifting the conformational landscape to influence substrate turnover hints to the numerous possibilities for regulatory fine-tuning of proteasomal degradation through posttranslational modifications or binding partners, such as Ubp6 (*Bashore et al., 2015*; *Aufderheide et al., 2015*). There is already a growing number of factors and site-specific modifications of the base and lid subcomplexes that are known to affect proteasome activities and are coupled to conformational switching (*VerPlank and Goldberg, 2017*; *VerPlank et al., 2019*). Furthermore, modulating the conformational equilibria through proteasome-interacting effectors could differentially influence the turnover of only specific substrate pools in the cell, as illustrated by our observations that the Rpn5-VTENKIF and Rpt6-EQ mutations led to a range of degradation defects depending on the substrate identity. Given the extensive structural and functional conservation of 26S proteasomes between yeast and humans (*Bard et al., 2018a*; *Finley et al., 2016*; *Kachroo et al., 2015*), we expect this mechanism of conformational regulation to be conserved among eukaryotic proteasomes.

# Materials and methods

**Key resources table**

| Reagent type (species) or resource | Designation | Source or reference | Identifiers | Additional information |
| --- | --- | --- | --- | --- |
| Recombinant DNA reagent | pET-Duet Rpn1, Rpn2, Rpn13 | (*Beckwith et al., 2013*) | pAM81 | |
| Recombinant DNA reagent | pACYC-Duet RIL Nas6, Hsm3, Rpn14, Nas2 | (*Beckwith et al., 2013*) | pAM83 | |
| Recombinant DNA reagent | pCOLA FLAG-Rpt1, Rpt2, His$_6$-Rpt3, Rpt5, Rpt6, Rpt4 | (*Beckwith et al., 2013*) | pAM82 | |

*Continued on next page*

*Continued*

| Reagent type (species) or resource | Designation | Source or reference | Identifiers | Additional information |
|---|---|---|---|---|
| Recombinant DNA reagent | pCOLA FLAG-Rpt1-191TAG, Rpt2, His$_6$-Rpt3, Rpt5, Rpt6, Rpt4 | (*Bard et al., 2019*) | pAM88 | |
| Recombinant DNA reagent | Synthetase pEVOL mod. | (*Bard et al., 2019*) (*Worden et al., 2017*) | pAM183 | |
| Recombinant DNA reagent | pCOLA FLAG-Rpt1, sspB permutant-Rpt2, His$_6$-Rpt3, Rpt5, Rpt6, Rpt4 | (*Bashore et al., 2015*) | pAM210 | |
| Recombinant DNA reagent | pCOLA FLAG-Rpt1-EQ, Rpt2, His$_6$-Rpt3, Rpt5, Rpt6, Rpt4 | (*Beckwith et al., 2013*) | pAM204 | |
| Recombinant DNA reagent | pCOLA FLAG-Rpt1, Rpt2-EQ, His$_6$-Rpt3, Rpt5, Rpt6, Rpt4 | (*Beckwith et al., 2013*) | pAM205 | |
| Recombinant DNA reagent | pCOLA FLAG-Rpt1, Rpt2, His$_6$-Rpt3-EQ, Rpt5, Rpt6, Rpt4 | (*Beckwith et al., 2013*) | pAM209 | |
| Recombinant DNA reagent | pCOLA FLAG-Rpt1, Rpt2, His$_6$-Rpt3, Rpt5, Rpt6, Rpt4-EQ | (*Beckwith et al., 2013*) | pAM206 | |
| Recombinant DNA reagent | pCOLA FLAG-Rpt1, Rpt2, His$_6$-Rpt3, Rpt5-EQ, Rpt6, Rpt4 | (*Beckwith et al., 2013*) | pAM207 | |
| Recombinant DNA reagent | pCOLA FLAG-Rpt1, Rpt2, His$_6$-Rpt3, Rpt5, Rpt6-EQ, Rpt4 | (*Beckwith et al., 2013*) | pAM208 | |
| Recombinant DNA reagent | pCOLA MBP-HRV3C-Rpt1, Rpt2, His$_6$-HRV3C-Rpt3, Rpt5, Rpt6-EQ, Rpt4 | This study | pAM214 | This plasmid encodes HRV3C cleavable affinity tags to make tagless recombinant base. |
| Recombinant DNA reagent | His$_6$-Ubp6 | (*Bashore et al., 2015*) | pAM211 | |
| Recombinant DNA reagent | His$_6$-Ubp6 C118A | (*Bashore et al., 2015*) | pAM212 | |
| Recombinant DNA reagent | MGCS-titin I27 $^{V15P}$ (lysineless)-ssrA-1K-35 amino acid tail including PPPY and His$_6$ | (*de la Peña et al., 2018*) | pAM213 | |
| Recombinant DNA reagent | titin I27$^{V13P/V15P}$(lysineless)-PPPY-ssrA-1K-35 amino acid tail containing ssrA | (*Bard et al., 2019*) | pAM94 | |
| Recombinant DNA reagent | titin I27$^{V15P}$(lysineless)-PPPY-ssrA-1K-35 amino acid tail | (*Bard et al., 2019*) | pAM91 | |
| Recombinant DNA reagent | titin I27 (lysineless)-PPPY-ssrA-1K-35 amino acid tail | (*Bard et al., 2019*) | pAM93 | |
| Recombinant DNA reagent | His$_6$-thrombin-N1-G3P (lysineless)—1 K-54 amino acid tail including ssrA, PPPY, C-terminal lysineless StrepII tag. | (*Myers et al., 2018*) | pAM77 | |
| Recombinant DNA reagent | His$_6$-SUMO-Ub$_4$ | (*Bard et al., 2019*) | pAM102 | |

*Continued on next page*

*Continued*

| Reagent type (species) or resource | Designation | Source or reference | Identifiers | Additional information |
|---|---|---|---|---|
| Recombinant DNA reagent | pET lid wild-type (Rpn5, MBP-HRV3C-Rpn6, Rpn8, Rpn11, Rpn9) | (*Bard et al., 2019*) | pAM85 | |
| Recombinant DNA reagent | pET lid VTENKIF (Rpn5-VTENKIF, MBP-HRV3C-Rpn6, Rpn8, Rpn11, Rpn9) | This study | pAM203 | This plasmid encodes a cleavable MBP tag on Rpn6 and is used to make tagless Rpn5-VTENKIF lid. |
| Recombinant DNA reagent | pCOLA (His$_6$-HRV3C-Rpn12, Rpn7, Rpn3) | (*Bard et al., 2019*) | pAM86 | |
| Recombinant DNA reagent | pACYC Sem1, Hsp90 | (*Lander et al., 2012*) | pAM80 | |
| Recombinant DNA reagent | pRS305 His$_{10}$-HRV3C-RPN5 | This study | pAM198 | This plasmid encodes an *S. cerevisiae* integratable Rpn5 gene with endogenous promotors and a cleavable N-terminal histidine tag on Rpn5. |
| Recombinant DNA reagent | pRS305 His$_{10}$-HRV3C-rpn5-vtenkif-aaaaaaa | This study | pAM199 | This plasmid encodes an *S. cerevisiae* integratable Rpn5-VTENKIF gene with endogenous promotors and a cleavable N-terminal histidine tag on Rpn5. |
| Recombinant DNA reagent | pRS305 3 × FLAG-HRV3C-RPN5 | This study | pAM200 | This plasmid encodes an *S. cerevisiae* integratable Rpn5 gene with endogenous promotors and a cleavable N-terminal FLAG tag on Rpn5. |
| Recombinant DNA reagent | pRS305 3 × FLAG-HRV3C-rpn5-vtenkif-aaaaaaa | This study | pAM201 | This plasmid encodes an *S. cerevisiae* integratable Rpn5-VTENKIF gene with endogenous promotors and a cleavable N-terminal FLAG tag on Rpn5. |
| Recombinant DNA reagent | pRS316 RPN6 promoter-RPN5-RPN5 terminator | This study | pAM202 | This plasmid encodes the Rpn5 ORF with a Rpn6 promotor and Rpn5 terminator on an *S. cerevisiae* counter-selectable, non-integrating plasmid. |
| Strain, strain background *E. coli* | BL21(DE3) | Thermofisher | Cat#C601003 | |

*Continued on next page*

*Continued*

| Reagent type (species) or resource | Designation | Source or reference | Identifiers | Additional information |
|---|---|---|---|---|
| Strain, strain background *S. cerevisiae* | MATa ade2-1 his3-11,15 leu2-3,112 trp1-1 ura3-1 can1-100 bar1 PRE1::PRE1—3 × FLAG(KANMX6) | (*Beckwith et al., 2013*) | yAM54 | |
| Strain, strain background *S. cerevisiae* | MATa ade2-1, his3-11,15, LEU2::His$_{10}$-HRV3C-RPN5, trp1-1, ura3-1, can1-100, RPN11::RPN11-3XFLAG (HIS3) | This study | yAM99 | This strain bears pAM198 integrated at LEU2 in an 3X-FLAG Rpn11 background. |
| Strain, strain background *S. cerevisiae* | MATa ade2-1, his3-11,15, LEU2::His$_{10}$-HRV3C-rpn5-vtenkif-aaaaaaa, trp1-1, ura3-1, can1-100, RPN11::RPN11-3XFLAG (HIS3) | This study | yAM100 | This strain bears pAM199 integrated at LEU2 in an 3X-FLAG Rpn11 background. |
| Strain, strain background *S. cerevisiae* | MATa ade2-1 his3-11, 15,112 trp1-1 ura3-1 can1-100 bar1 rpn5Δ::NATMX6, pRS316-promoter-RPN6-RPN5-terminator-RPN6 | This study | yAM96 | This strain has endogenous Rpn5 deleted and replaced with NATMX6 with pAM202 as a covering plasmid in a W303 background. |
| Strain, strain background *S. cerevisiae* | MATa ade2-1 his3-11, 15 trp1-1 ura3-1 can1-100 bar1 rpn5Δ::NATMX6, LEU2::3 × FLAG-HRV3C-RPN5 | This study | yAM97 | This strain bears pAM200 integrated at LEU2 in a yAM96 background. |
| Strain, strain background *S. cerevisiae* | MATa ade2-1 his3-11, 15 trp1-1 ura3-1 can1-100 bar1 rpn5Δ::NATMX6, LEU2::3 × FLAG-HRV3C-rpn5-vtenkif-aaaaaaa | This study | yAM98 | This strain bears pAM201 integrated at LEU2 in a yAM96 background. |
| Antibody | Polyclonal rabbit anti-Rpn5 | Abcam | Cat#ab79773 | Dilution (1:5000) |
| Antibody | Polyclonal rabbit anti-Nas6 | Abcam | Cat#ab91447 | Dilution (1:5000) |
| Antibody | Monoclonal Goat anti-rabbit IgG-HRP | Bio-Rad | 170–6515 | Dilution (1:10000) |
| Peptide, recombinant protein | Fluorescein-HHHHHHLPETGG | Genscript | Custom ordered | |
| Peptide, recombinant protein | Bovine Serum Albumin | Sigma Aldrich | Cat#A9418 | |
| Software | UCSF Chimera | UCSF | https://www.cgl.ucsf.edu/chimera/ | |
| Software | Origin Pro | Origin Lab | https://www.originlab.com/ | |
| Software | ImageQuant | GE | ImageQuant TL 8.1 | |
| Chemical compound | Cy3 DBCO | Click Chemistry Tools | Cat#A140 | |
| Chemical compound | Fluorescein-5-maleimide | ThermoFisher | Cat#62245 | |
| Chemical compound | Cy5 Maleimide | Lumiprobe | Cat#23380 | |
| Chemical compound | 4-azido-L-phenylalanine | Amatek Chemical | Cat#A-7137 | |

*Continued on next page*

*Continued*

| Reagent type (species) or resource | Designation | Source or reference | Identifiers | Additional information |
|---|---|---|---|---|
| Chemical compound | 1,10-phenanthroline | Sigma Aldrich | Cat#P9375 | |

## Strain construction

Strains yAM96, yAM97, and yAM98 were constructed using standard techniques. A W303-derived parental strain was transformed with a pRS316-RPN5 (pAM202) covering plasmid and then transformed with a PCR product containing homologous regions flanking the RPN5 gene and containing the NATMX marker (*Longtine et al., 1998*; *Goldstein and McCusker, 1999*). RPN5 disruption was confirmed by PCR and sequencing of both the 5' and 3' junctions of the NATMX integration. RPN5 and rpn5-vtenkif-aaaaaaa were introduced by integration of pRS305 vectors containing promoter and terminator from Rpn5 that had been linearized in the LEU2 marker. Curing of the covering plasmid was performed twice sequentially on plates containing 5-FOA and confirmed by loss of growth on dropout URA plates. yAM99 and yAM100 strains were also constructed using standard techniques. The YYS40 (*Sone et al., 2004*) parental strain bearing RPN11::3X-FLAG-RPN11 (HIS3) was transformed with pRS305 linearized at the LEU2 marker and containing either 10X-Histag-HRV3C-RPN5 or 10X-Histag-HRV3C-rpn5-vtenkif-aaaaaaa. Integration was confirmed by PCR.

## Protein purification

### Purification of the tagged heterologous base and SspB-fused base

Preparation of the *Saccharomyces cerevisiae* base subcomplex was conducted as described previously (*Beckwith et al., 2013*; *Bashore et al., 2015*; *Worden et al., 2017*; *Bard and Martin, 2018b*). BL21-star (DE3) *E. coli* cells were transformed and grown in 3L of terrific broth, shaking at 37°C until $OD_{600}$ ~0.8–1.0 was reached. Temperature was lowered to 30°C and protein expression was induced with 1 mM IPTG for 5 hr at 30°C, followed by overnight expression at 16°C. Cells were harvested by centrifugation and resuspended in base lysis buffer (60 mM HEPES pH 7.6, 50 mM NaCl, 50 mM KCl, 10 mM $MgCl_2$, 5% glycerol, 2 mM ATP, + 2 mg/mL lysozyme, proteasome inhibitors (PMSF, Aprotonin, Leupeptin, PepstainA), and benzonase, and then stored at −80°C. For the purification, cells were thawed and lysed by sonication. Lysate was clarified by centrifugation and loaded onto a HisTrap High-Performance 5 mL columns (GE Healthcare) using a peristaltic pump, washed with base NiA buffer (60 mM HEPES pH 7.6, 50 mM NaCl, 50 mM KCl, 10 mM $MgCl_2$, 5% glycerol, 2 mM ATP + 20 mM imidazole), and eluted with base NiB buffer (60 mM HEPES pH 7.6, 50 mM NaCl, 50 mM KCl, 10 mM $MgCl_2$, 5% glycerol, 2 mM ATP, 250 mM imidazole). Eluates were then flowed over M2 ANTI-FLAG affinity resin (Sigma) and eluted with 0.5 mg/mL 3X FLAG peptide (Genscript) in base lysis buffer. Base subcomplex was further purified by size-exclusion chromatography using a Superose 6 increase 10/300 column (GE Healthcare) pre-equilibrated with base GF buffer (60 mM HEPES pH 7.6, 50 mM NaCl, 50 mM KCl, 10 mM $MgCl_2$, 5% glycerol, 0.5 mM TCEP, 1 mM ATP). Peak fractions corresponding to assembled base subcomplex were concentrated, flash frozen in liquid nitrogen, and stored at −80°C. The concentration of base was determined by Bradford protein assay using bovine serum albumin (BSA) as a standard.

### Purification and labeling of base containing unnatural amino acid

Preparation of 4-azidophenylalanine-containing base subcomplex was conducted as detailed previously (*Bard et al., 2019*; *Bard and Martin, 2018b*). BL21star (DE3) *E. coli* were cultured overnight in 2xYT media and diluted into prewarmed media containing antibiotics (300 µg/mL Ampicillin, 25 µg/mL Chloramphenicol, 50 µg/mL kanamycin, and 100 µg/mL spectinomycin). Cells were grown with shaking to $OD_{600} = 0.6$ before pelleting and resuspending, pooling 6L of cells into 1L of buffered TB containing 2 mM 4-azidophenylalanine, 17 mM $KH_2PO_4$, and 72 mM $K_2HPO_4$ at 30°C. After 30 min, protein expression was induced with 1 mM IPTG for 5 hr, followed by overnight incubation with shaking at 16 °C.

Following centrifugation, cells were resuspended in base lysis buffer, and purification was performed as described above for heterologously expressed base until elution from FLAG affinity column. After elution from M2 ANTI-FLAG affinity resin (Sigma), artificial amino acid-containing base

was incubated at room temperature with 150 μM 5,5′-dithiobis-2-nitrobenzoic acid for 10 min before chilling on ice and adding 300 μM DBCO-Cy3 (Click Chemistry Tools) and incubating at 4°C overnight. Following overnight labeling, the reaction was quenched with 10 mM DTT and subjected to size-exclusion chromatography on a Superose 6 Increase 10/300 (GE Healthcare) in GF buffer, as described for other base constructs above. Base concentration was determined by Bradford protein assay using BSA as a standard, while the extent of Cy3 labeling was determined by absorbance at 555 nm, and SDS-PAGE was used to confirm labeling of only Rpt1 as well as complete removal of free dye.

## Purification of the heterologously expressed tagless base subcomplex

Preparation of recombinantly expressed, tagless *S. cerevisiae* base subcomplex was conducted using standard affinity-chromatography and size-exclusion chromatography protocols. Briefly, BL21-star (DE3) *E. coli* cells were grown in 3L of terrific broth shaking at 37°C until $OD_{600}$ ~0.8–1.0 was reached. Temperature was lowered to 30°C and protein expression was induced with 1 mM IPTG for 5 hr at 30°C, followed by overnight expression at 16°C. Cells were harvested by centrifugation and resuspended in base lysis buffer (60 mM HEPES pH 7.6, 50 mM NaCl, 50 mM KCl, 10 mM $MgCl_2$, 5% glycerol, 2 mM ATP, + 2 mg/mL lysozyme, proteasome inhibitors (PMSF, Aprotonin, Leupeptin, PepstainA), and benzonase, and then stored at −80°C. For the purification, cells were thawed and lysed by sonication. Lysate was clarified by centrifugation and loaded onto HisTrap High-Performance 5 mL (GE Healthcare) columns using a peristaltic pump, washed with base NiA buffer (60 mM HEPES pH 7.6, 50 mM NaCl, 50 mM KCl, 10 mM $MgCl_2$, 5% glycerol, 2 mM ATP + 20 mM imidazole), and eluted with base NiB buffer (60 mM HEPES pH 7.6, 50 mM NaCl, 50 mM KCl, 10 mM $MgCl_2$, 5% glycerol, 2 mM ATP, 250 mM imidazole). Eluates were flowed over Amylose Resin (NEB), washed with base GF buffer (60 mM HEPES pH 7.6, 50 mM NaCl, 50 mM KCl, 10 mM $MgCl_2$, 5% glycerol, 0.5 mM TCEP, 2 mM ATP), and eluted with base GF buffer + 10 mM maltose + ATP regeneration system (creatine kinase and creatine phosphate). HRV3C protease was added to the Amylose eluate and cleavage was allowed to proceed for 45 min at room temperature or overnight at 4°C. The Amylose resin eluate was concentrated and loaded onto a Superose 6 increase 10/300 size exclusion column equilibrated with base GF buffer. Peak fractions corresponding to assembled base were concentrated, flash frozen, and stored at −80°C. The concentration of base was determined by Bradford protein assay using BSA as a standard.

## Purification of Rpn10, core particle, Ubp6, Ubp6 C118A, *M. musculus* Uba1, *S. cerevisiae* Ubc4, *S. cerevisiae* Rsp5, ubiquitin, and linear ubiquitin tetramer

Rpn10, core particle, *M. musculus* Uba1, Ubc4, Rsp5, and ubiquitin were prepared as described in *Worden et al. (2017)* using standard expression and purification procedures (*Bashore et al., 2015*; *Worden et al., 2017*; *Bard and Martin, 2018b*). Purification of Ubp6 and Ubp6-C118A was performed as described (*Bashore et al., 2015*), and linear ubiquitin tetramer was purified exactly as described (*Bard et al., 2019*).

## Purification of His$_{10}$-HRV3C-Rpn5-VTENKIF mutant and wild-type 26S holoenzymes

Yeast strains yAM99 (wild type Rpn5) and yAM100 (mutant Rpn5) were grown in 3L of YPD for 3 days at 30°C. Cells were harvested by centrifugation, weighed, and resuspended in 15 mL of 26S lysis buffer (60 mM HEPES pH 7.6, 25 mM NaCl, 10 mM $MgCl_2$, 2.5% glycerol, 5 mM ATP + ATP regeneration (creatine kinase and creatine phosphate)). Resuspended cells were flash frozen in liquid nitrogen, lysed by cryo grinding, and stored at −80°C. Lysed yeast powder was thawed at room temperature and diluted in 26S lysis buffer to 1.5 mL buffer per gram of yeast. Lysate was clarified by centrifugation and bound in batch to M2 ANTI-FLAG affinity resin (Sigma) for 1 hr at 4°C. FLAG resin was subsequently washed in batch twice with 25 mL of 26S lysis buffer, applied to a gravity flow column, and washed with an additional 25 mL of 26S lysis buffer. Proteasome was eluted with 26S lysis buffer + 0.5 mg/mL 3X FLAG peptide. FLAG eluate was loaded onto a 1 mL HisTrap High-Performance 5 mL columns (GE Healthcare) using a peristaltic bump and washed with five column volumes of 26S NiA buffer (30 mM HEPES pH 7.6, 10 mM $MgCl_2$, 10% glycerol, 5 mM ATP, 10 mM

imidazole). Proteasome was eluted with 26S NiB buffer (30 mM HEPES pH 7.6, 10 mM $MgCl_2$, 10% glycerol, 5 mM ATP, 500 mM imidazole). HRV3C protease was added in excess, and cleavage was allowed to proceed for 30 min at 4°C. 26S proteasome was concentrated and loaded onto a Superose 6 increase 10/300 size exclusion column pre-equilibrated with 26S GF buffer (60 mM HEPES pH 7.6, 25 mM NaCl, 10 mM $MgCl_2$, 2.5% glycerol, 1 mM ATP, 0.5 mM TCEP). Peak fractions were spiked with ATP regeneration (creatine kinase and creatine phosphate), concentrated, flash frozen in liquid nitrogen, and stored at 80°C. 26S holoenzyme concentration was determined by Bradford protein assay using BSA as a standard for total protein and in-gel quantification using purified Rpn1 as an internal standard for total regulatory particle.

## Purification of FLAG-HRV3C-Rpn5 mutant and wild-type 26S holoenzymes

Yeast strains yAM97 (wild type Rpn5) and yAM98 (mutant Rpn5) were grown in 3L of YPD for 3 days at 30°C. Cells were harvested by centrifugation, weighed, and resuspended in 15 mL of 26S lysis buffer (60 mM HEPES pH 7.6, 25 mM NaCl, 10 mM $MgCl_2$, 2.5% glycerol, 5 mM ATP + ATP regeneration (creatine kinase and creatine phosphate)). Resuspended cells were flash frozen in liquid nitrogen, lysed by cryo grinding (SPEX Freezer/Mill), and stored at −80°C. Lysed yeast powder was thawed at room temperature and diluted in 26S lysis buffer to 1.5 mL buffer per gram of yeast. Lysate was clarified by centrifugation and bound in batch to M2 anti-FLAG affinity resin (Sigma) for 1 hr at 4°C. FLAG resin was subsequently washed in batch twice with 25 mL of 26S lysis buffer, applied to a gravity flow column, and washed with an additional 25 mL of 26S lysis buffer. 26S proteasome was eluted with 26S lysis buffer + 0.5 mg/mL 3X FLAG peptide (Genscript). FLAG eluate was cleaved with HRV protease added in excess, and cleavage was allowed to proceed for 30 min at 4°C. 26S proteasome was concentrated and loaded onto a Superose 6 increase 10/300 size-exclusion column pre-equilibrated with 26S GF buffer (60 mM HEPES pH 7.6, 25 mM NaCl, 10 mM $MgCl_2$, 2.5% glycerol, 1 mM ATP, 0.5 mM TCEP). Peak fractions were spiked with ATP regeneration system (creatine kinase and creatine phosphate), concentrated, flash frozen in liquid nitrogen, and stored at −80°C. 26S holoenzyme concentration was determined by Bradford protein assay using BSA as a standard for total protein and in-gel quantification using purified Rpn1 as a standard for total regulatory particle.

## Purification of the heterologous lid

Heterologous expression and purification of the *Saccharomyces cerevisiae* lid subcomplex was performed similarly to previous studies (*Bard et al., 2019*). BL21-star (DE3) *E. coli* cells were grown in 2L of terrific broth medium shaking at 37°C until an $OD_{600}$ ~1.0–1.5 was achieved. Protein expression was induced with 1 mM IPTG overnight at 18°C. Cells were harvested by centrifugation and resuspended in lid lysis buffer (60 mM HEPES pH 7.6, 25 mM NaCl, 10 mM $MgCl_2$, 2.5% glycerol + 2 mg/mL lysozyme, proteasome inhibitors (PMSF, Aprotonin, Leupeptin, PepstainA), and benzonase, and then stored at −80°C. For purification, cells were thawed and lysed by sonication. Lystate was clarified by centrifugation and loaded onto HisTrap High-Performance 5 mL columns (GE Healthcare) using a peristaltic pump, washed with lid NiA buffer (60 mM HEPES pH 7.6, 25 mM NaCl, 10 mM $MgCl_2$, 2.5% glycerol + 20 mM imidazole), and eluted with lid NiB buffer (60 mM HEPES pH 7.6, 25 mM NaCl, 10 mM $MgCl_2$, 2.5% glycerol, 250 mM imidazole). Eluates were flowed over Amylose Resin (NEB), washed with lid GF buffer (60 mM HEPES pH 7.6, 25 mM NaCl, 10 mM $MgCl_2$, 2.5% glycerol, 0.5 mM TCEP), and eluted with lid GF buffer + 10 mM maltose. HRV3C protease was added to the Amylose eluate and cleavage was allowed to proceed overnight at 4°C or at room temperature for 2 hr. Amylose resin eluate was concentrated and loaded onto a Superose 6 increase 10/300 size-exclusion column equilibrated with GF buffer. Peak fractions corresponding to assembled lid were concentrated, flash frozen, and stored at −80°C.

## Substrate preparation and ubiquitination

G3P substrate preparation and labeling was performed as described previously (*Worden et al., 2017*), and the titin-I27$^{V13P/V15P}$, titin-I27$^{V15P}$, and titin-I27 substrates were purified and labeled as described (*Bard et al., 2019*; *de la Peña et al., 2018*). Ubiquitination reactions were carried out as described previously (*Bard et al., 2019*; *de la Peña et al., 2018*; *Myers et al., 2018*). Briefly, 10–20

µM substrate protein was incubated with 2 µM mouse E1 enzyme (mE1), 5 µM Ubc4, and 5 µM Rsp5 with 450–800 µM ubiquitin and 6–10 mM ATP in 25 mM HEPES pH 8.0, 150 mM NaCl, 5% glycerol at 25℃ until completion (assessed by SDS-PAGE, 30–180 min). Ubiquitination reaction conditions were screened for uniform higher molecular weights and full non-ubiquitinated substrate depletion by SDS-PAGE.

## Native polyacrylamide gel electrophoresis of purified assembled proteasomes

Proteasomes were reconstituted with 1 µM core particle and 2 µM base, lid, and Rpn10 in GF buffer with 5 mM ATP, 0.5 mM TCEP, and ATP regeneration system, and allowed to assemble for 5 min at room temperature. Equivalent amounts of reconstituted proteasomes were diluted appropriately in 5X native gel sample buffer (250 mM Tris*HCl pH 7.5, 50 mM $MgCl_2$, 1 mM ATP, 50% glycerol, 0.015% w/v xylene cyanol) and loaded onto 4% native polyacrylamide gels with 1 mM ATP and a 3% polyacrylamide stacking gel containing 2.5% sucrose and 1 mM ATP. Samples were electrophoresed at 100 V and 4℃ for 4 hr as described (Elsasser et al., 2004). In-gel peptidase activity was assayed by incubating the gel in GF buffer with 5 mM ATP, 0.5 mM TCEP, and 100 µM Suc-LLVY-AMC with or without 0.02% SDS for 10 min before imaging on a Chemidoc MP Imaging System (Bio-Rad). The same gel was subsequently fixed and Coomassie stained for detection of total protein. Where indicated, samples from reconstitutions were further diluted in 2X SDS-PAGE sample buffer, electrophoresed under denaturing conditions, and imaged as a loading control.

## Negative-stain transmission electron microscopy

Wild-type *S. cerevisiae* 26S holoenzyme was diluted to ~400 nM in a buffer (60 mM HEPES, pH 7.6, 25 mM NaCl, 10 mM MgCl2, 1 mM TCEP) supplemented with 6 mM ATP or 2 mM ATPγS. 4 µL of the ATP- or ATPγS-containing solution were applied to a plasma treated (Electron Microscopy Sciences) carbon film supported by a Maxtaform 400 mesh Cu/Rh grid (TED PELLA). After incubation for 45 s, excess solution was wicked with Whatman #1 filter paper and immediately treated with a 2% (w/v) solution of uranyl formate stain. Excess stain was removed by wicking, and the grids were allowed to dry for 10 min before visualization by transmission electron microscopy. The same dilution, blotting, and staining approach was used for a solution containing Rpn5-VTENKIF-mutant 26S holoenzyme purified as described above.

Data were acquired with the Leginon automation software and a Tecnai F20 transmission electron microscope (FEI) operated at 200 keV with an under-focus range of 0.5–1.0 µm. A total fluence of 30 $e^-/Å^2$ was used to collect ~800 micrographs for each of the 26S holoenzyme variants in ATP or ATPγS with an Eagle 4 k CCD camera (FEI) at a nominal magnification of 62,000x and amplified pixel size of 1.79 Å. Approximately 800 micrographs were processed for each of the four datasets using single particle analysis (SPA) with RELION 3.0b3. The extracted particles were subject to the same SPA workflow (*Figure 2—figure supplement 1B*) with a final 3D classification step into six classes to quantify the degree of heterogeneity present in each dataset (*Figure 2—figure supplement 1–6*, *Figure 2A*). Proteasome conformational state for each class was determined for each state using UCSF Chimera's 'Fit in Map' tool comparing each class to the atomic models in *Eisele et al. (2018)*.

## Anti-FLAG pulldown of assembled proteasome complexes

Proteasomes were reconstituted with 500 nM core particle and 1 µM tagless base, lid, and Rpn10, and allowed to assemble for 5 min in GF buffer with 1 mg/mL BSA, 5 mM ATP, and ATP regeneration system at room temperature. Magnetic ANTI-FLAG m2 resin (Sigma) was added to the solution and resin binding was allowed to proceed at 4℃ for 1 hr. Resin was washed three times with 120 µL of GF buffer including 1 mg/mL BSA and 5 mM ATP, before eluting bound complexes with 35 µL of GF buffer supplemented with 5 mM ATP and 1 mg/mL 3X FLAG peptide at 30℃ for 30 min.

## Immunoblot analysis

SDS-PAGE gels including Precision Plus Stained protein standards (Thermo Fisher) were transferred to activated 0.2 µm PVDF membrane (Thermo Scientific) via semi-dry transfer in (25 mM Tris-HCl pH 8.3, 192 mM glycine, 5% methanol) for 45 min using constant 80 mA current before membrane blocking with 5% milk in TBST (50 mM Tris-HCl pH 7.6, 150 mM NaCl, 0.05% Tween-20) for at least

one hour. Blocked membranes were probed with primary antibody (diluted in TBST with 5% milk) for at least 1 hr before being washed with TBST and re-probed with secondary anti-Rabbit-HRP for at least 30 min. Membranes were subsequently washed three times with TBST (15 min each) before visualization of Chemiluminescence activity using Western Lightning ECL reagent (Perkin Elmer) in Chemidoc MP Imaging System (Bio-Rad) with exposure times ranging from 30 to 120 s.

## ATPase activity measurements

ATP-hydrolysis rates were determined using an NADH-coupled assay (pyruvate kinase and lactate dehydrogenase) as described previously (Beckwith et al., 2013; Bashore et al., 2015). Briefly, proteasomes were reconstituted under base-limiting conditions with 200 nM base of the indicated mutant, 800 nM core, 800 nM lid, and 1 µM Rpn10 in GF buffer with 5 mM ATP and 0.5 mM TCEP at room temperature for 5 min, before being 2-fold diluted into ATPase mix (final concentrations: 1 mM NADH, 5 mM ATP, 7.5 mM phosphoenolpyruvate, 3 U/mL pyruvate kinase, and 3 U/mL lactate dehydrogenase), applied to a 384-clear bottom plate (Corning), and centrifuged at (1000 x g) for 1 min prior to measurement. Steady-state depletion of NADH was assessed by measuring the absorbance at 340 nm in a Synergy Neo2 Multi-Mode Plate Reader (Biotek). Solution pathlength was manually determined per experiment through titration of NADH and used to calculate ATPase rate.

For ATPase response to ubiquitin-bound Ubp6, measurements were performed as described above, using Rpn10-ΔUIM rather than full length Rpn10 and with the addition of 400 nM Ubp6-C118A and 100 µM linear $Ub_4$.

To determine lid affinity through the ATPase response of the base, holoenzymes were assembled at room temperature for 5 min with 100 nM base, 1.6 µM core, 4 µM Rpn10, and varying concentrations of lid, before a 2-fold dilution with ATPase mix to start the reaction. For measurements in the presence of substrate, ubiquitinated titin-I27$^{V15P}$ was added at a final concentration of 3 µM.

## Ubp6 Ub-AMC cleavage-activity assays

Ubp6 activity was measured using the cleavage of the fluorogenic Ub-AMC substrate (Life Sensors). Proteasomes were reconstituted as base-limited complexes (at a final concentration of 200 nM base, 1.2 µM lid, 600 µM core, 1.5 µM Rpn10ΔUIM) in either 1X ATP regeneration mix or 4 mM ATPγS with 40 nM Ubp6. After 4 min preincubation at 25 °C, samples were mixed with Ub-AMC to a final concentration of 10 µM. Cleavage was measured by monitoring the change of fluorescence at 445 nm after excitation at 345 nm on a plate reader (Synergy Neo2 Multi-Mode Plate Reader, Biotek).

## Measurement of peptidase stimulation

Proteasomes were reconstituted at 2X final concentration with limiting concentration of core particle (10 nM final) and saturating concentrations of base (0.5 µM (1X) or 1 µM (2X) for experiments where the base concentration was doubled), lid (2 µM), and Rpn10 (2 µM) in GF buffer supplemented with 0.5 mM TCEP and 5 mM ATP for 5 min at room temperature. Reconstituted proteasomes were incubated in either 5 mM ATP or 5 mM ATPγS at room temperature for an additional 5 min. Suc-LLVY-AMC was diluted to 2X concentration (100 µM final) in 26S GF buffer. Reactions were initiated by aliquoting 5 µL of reconstituted proteasomes into 5 µL of Suc-LLVY-AMC solution in a 384-well flat bottom black corning plate. Suc-LLVY-AMC hydrolysis was tracked by the increase in fluorescence upon AMC release in a Synergy Neo2 Multi-Mode Plate Reader (Biotek). Data were fit by linear regression, and slopes were normalized to wild-type proteasomes in ATP.

## Proteasome degradation assays

### Michaelis-Menten analyses of titin substrate degradation monitored by fluorescence anisotropy

Proteasomes were reconstituted at 2X concentration with limiting concentrations of core particle (100 nM final) and saturating concentrations of base, lid, and Rpn10 (2 µM final) for 5 min at room temperature in assay buffer (GF buffer supplemented with 5 mM ATP, 1 mg/mL BSA, and ATP regeneration (creatine kinase and creatine phosphate)). Fluorescein labeled titin-I27 with a V15P mutation and a C-terminal 35 residue tail (FAM-titin-I27$^{V15P}$) was prepared at 2X final concentration in assay buffer. Reactions were initiated with 5 µL of proteasome sample being added to 5 µL of FAM-titin-I27$^{V15P}$ substrate in a 384-well flat bottom black corning plate. Degradation was

monitored by the loss of fluorescence anisotropy of conjugated fluorescein over time in a Synergy Neo2 Multi-Mode Plate Reader (Biotek). Degradation rates were calculated by determining the fluorescence anisotropy difference between substrate and substrate peptides and applying linear regression to initial anisotropy decreases. Initial rates were plotted against substrate concentration and fitted to the Michaelis-Menten equation (OriginPro9) to determine $k_{cat}$ and $K_m$ values.

## Multiple-turnover degradation measured by fluorescence anisotropy

Proteasomes were either reconstituted at 2X concentration with limiting concentrations of core particle (100 nM final) and saturating concentrations of base (0.5 µM), lid (2 µM), Rpn10 (2 µM final) for 5 min at room temperature or purified holoenzyme was diluted to 100 nM (final) in 26S GF buffer with 5 mM ATP, 1 mg/mL BSA, and an ATP regeneration system (creatine kinase and creatine phosphate). Substrate was prepared at 2X concentration in 26S GF buffer. Reactions were initiated with 5 µL of proteasome sample being added to 5 µL of substrate in a 384-well flat bottom black corning plate. Degradation was monitored by the loss of fluorescence anisotropy in a Synergy Neo2 Multi-Mode Plate Reader (Biotek). Degradation rates were calculated by determining the fluorescence anisotropy difference between undegraded substrate and fully degraded substrate (using chymotrypsin (Sigma) to fully degrade substrate) and linear regression.

## Single-turnover degradation measured by fluorescence anisotropy

Proteasomes were either reconstituted at 2X concentration with limiting concentrations of core particle (0.9 µM final) and saturating concentrations of base, lid, and Rpn10 (2.5 µM each) for 5 min at room temperature or purified holoenzyme was diluted to 2X concentration (2 µM final) in 26S GF buffer with 5 mM ATP, 1 mg/mL BSA, and an ATP regeneration system (creatine kinase and creatine phosphate). Substrate was prepared at 2X concentration (150 nM final) in GF buffer. Reactions were initiated with 2.5–5 µL of proteasome sample being added to 2.5–5 µL of substrate in a 384-well flat bottom black Corning plate. Degradation was monitored by loss of fluorescence anisotropy in a Synergy Neo2 Multi-Mode Plate Reader (Biotek). Degradation rates were calculated by fitting fluorescence anisotropy traces to a double exponential decay model, see *Equation 1* below (OriginPro9).

To assess the effects of doubling the concentration of Rpn5-VTENKIF proteasome, holoenzyme was reconstituted at 4X concentration with limiting concentration of core particle (0.9 µM final) and saturating concentrations of base, lid, and Rpn10 (2.5 µM each, final) for 5 min at room temperature in 26S GF buffer with 5 mM ATP, 1 mg/mL BSA, and an ATP regeneration system (creatine kinase and creatine phosphate). Proteasome was either kept undiluted or diluted to 2X concentration with GF buffer before reactions were initiated with 2.5 µL of 2X substrate (150 nM final) in a 384-well flat-bottom black Corning plate. Anisotropy change over time was observed as described above.

## Effects of the regulatory particle on substrate processing measured by fluorescence anisotropy

Regulatory particles were reconstituted at 4X concentration with equimolar base, lid, and Rpn10 (2.5 µM each, final at 1X) for 5 min at room temperature in GF buffer with 5 mM ATP, 1 mg/mL BSA, and an ATP regeneration system (creatine kinase and creatine phosphate), either alone or incubated with core particle (900 nM core particle final; 2.5 µM RP final at 1X or 5 µM RP final at 2X). Control with core particle alone were prepared by mixing 2X core particle (900 nM final) in GF buffer with 5 mM ATP, 1 mg/mL BSA, and an ATP regeneration system (creatine kinase and creatine phosphate). Reactions were initiated by adding 5 µL 2X substrate (150 nM final) to RP/proteasomes in a 384-well flat-bottom black Corning plate. Substrate processing was monitored by the decrease in fluorescence anisotropy in a Synergy Neo2 Multi-Mode Plate Reader (Biotek). After completion of the measurements, samples were diluted with 2X SDS-PAGE sample buffer for SDS-PAGE analysis.

## Single-turnover degradation monitored by SDS-PAGE

Gel-based single-turnover measurements of FAM-titin-I27 degradation were initiated as described above. 1.2 µL aliquots at various time points were quenched in 2X SDS-PAGE loading buffer (5 µL) and electrophoresed on 4–20% TGX SDS-PAGE gels (Bio-Rad). Gels were imaged on a Typhoon variable mode scanner (GE Healthcare) for fluorescein fluorescence. Gel lanes were quantified for fraction-of-total fluorescence intensity using ImageQuant (GE Healthcare).

## Substrate-tail insertion monitored by FRET

Similar to the previously described procedure (*Bard et al., 2019*), substrate-tail insertion was measured by detecting FRET between Cy5-labeled ubiquitinated FAM-titin-I27$^{V15P}$ substrate and Cy3-labeled, Rpn11-inhibited proteasomes under single-turnover conditions. Reactions containing 2-fold concentrated, base-limited, and *o*-PA-inhibited holoenzyme (220 nM base containing Rpt1-$^{I191AzF-Cy3}$, 1.2 µM lid, 800 nM core, 1.5 µM Rpn10, 6 mM *o*-PA and either 2X ATP Regeneration system or 2.5 mM ATPγS) were mixed with 2X concentrated ubiquitinated Cy5-labeled FAM-titin-I27$^{V15P}$ substrate (6 µM, as 2X stock) in an Auto SF120 stopped flow fluorometer (Kintek). Samples were excited at 550 nm with emission at 576 nm (Cy3) and 690 nm (Cy5) measured simultaneously. Kinetics were determined by fitting of the Cy5 gain of signal to *Equation 2*.

For substrate-tail insertion reactions monitored by FRET under single-turnover conditions, substrate was prepared as described above. Proteasomes were reconstituted at 2X concentration with limiting amounts of core particle (0.9 µM final) and saturating amounts of base, lid, and Rpn10 (2.5 µM each) for 5 min at room temperature in 26S GF buffer with 5 mM ATP, 1 mg/mL BSA, 6 mM *o*-PA, and an ATP regeneration system (creatine kinase and creatine phosphate). Substrate was prepared at 2X concentration (150 nM final) in GF buffer. Reactions were initiated with 2.5 µL of proteasome sample being added to 2.5 µL of substrate in a 384-well flat-bottom black Corning plate. FRET was monitored by simultaneous detection of Cy3 (680 nm, 30 nm bandpass filter) and Cy5 (590 nm, 35 nm bandpass filter) after excitation at 540 nm (25 nm bandpass) on a Synergy Neo2 Multi-Mode Plate Reader (Biotek).

## Proteasome restart assays

Assays were performed similarly as described (*Worden et al., 2017*). Briefly, proteasomes were reconstituted in GF buffer with 10 mM ATP and 0.5 mM DTT and allowed to assemble at 20℃ for 3 min. Single-turnover reactions were initiated with ubiquitinated TAMRA-G3P substrate. Under restart conditions, assembled proteasome were stalled with substrate by incubating with *o*-PA (3 mM final) for an additional 3 min at 20℃ before substrate addition. Stalled proteasomes were restarted by additional of GF with ZnCl$_2$ at a final concentration of 1 mM. From each reaction, 1.2 µL aliquots after various times were collected and quenched in 5 µL sample buffer (50 mM Tris pH 6.8, 20% glycerol, 0.2% SDS). Gel samples were electrophoresed on Criterion TGX 4–20% SDS-PAGE gels (Bio-Rad) and imaged on a Typhoon variable mode scanner (GE Healthcare) for TAMRA fluorescence using at least 25 µm per pixel resolution. Gels were quantified for fluorescence intensity using ImageQuant (GE Healthcare). Each lane was partitioned into segments for poly-ubiquitinated substrate (Ub$_n$), unmodified substrate, and peptide products and intensities were quantified as a fraction of total lane intensity. These data were plotted v time and fit to a first order exponential, see *Equation 3* below (OriginPro9) to derive degradation rates.

## Equations

Double Exponential Decay:

$$y = y_0 + A_1 {}^* \exp(-(x - x_0)/k_1) + A_2 {}^* \exp(-(x - x_0)/k_2)$$ (1)

Single exponential decay with linear component:

$$y = y_0 + A_1 {}^* \exp(-x/k_1) + m^* x$$ (2)

Single Exponential Decay:

$$y = y_0 + A \exp(-x/k)$$ (3)

## Acknowledgements

We thank the members of the Martin lab for helpful discussions, and particularly Jared Bard and Ken Dong for providing material. All EM data were collected at the Scripps Research (SR) electron microscopy facility. We thank B Anderson for microscope support and JC Ducom at SR High Performance Computing facility for computational support.

## Additional information

### Competing interests

Andreas Martin: Reviewing editor, *eLife*. The other authors declare that no competing interests exist.

### Funding

| Funder | Grant reference number | Author |
| --- | --- | --- |
| Howard Hughes Medical Institute | | Andreas Martin |
| National Institutes of Health | R01-GM094497 | Andreas Martin |
| National Institutes of Health | DP2EB020402 | Gabriel C Lander |
| American Cancer Society | 132279-PF-18-189-01-DMC | Andres H de la Peña |
| Pew Charitable Trusts | | Gabriel C Lander |
| National Institutes of Health | S10OD021634 | Gabriel C Lander |

The funders had no role in study design, data collection and interpretation, or the decision to submit the work for publication.

### Author contributions

Eric R Greene, Ellen A Goodall, Conceptualization, Resources, Formal analysis, Validation, Investigation, Writing - original draft, Writing - review and editing; Andres H de la Peña, Formal analysis, Investigation, Visualization, Writing - review and editing; Mary E Matyskiela, Conceptualization, Resources, Validation, Investigation; Gabriel C Lander, Supervision, Funding acquisition; Andreas Martin, Conceptualization, Supervision, Funding acquisition, Investigation, Methodology, Writing - original draft, Project administration, Writing - review and editing

### Author ORCIDs

Eric R Greene (iD) https://orcid.org/0000-0003-1717-0914
Ellen A Goodall (iD) http://orcid.org/0000-0001-9876-5973
Gabriel C Lander (iD) https://orcid.org/0000-0003-4921-1135
Andreas Martin (iD) http://orcid.org/0000-0003-0923-3284

### Decision letter and Author response

Decision letter https://doi.org/10.7554/eLife.49806.sa1
Author response https://doi.org/10.7554/eLife.49806.sa2

## Additional files

### Supplementary files

• Transparent reporting form

### Data availability

All data generated or analysed during this study are included in the manuscript and supporting files.

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
