## [Decision Letter]

**Acceptance summary:**

The 26S proteasome, a ~2 MDa complex of multiple subunits, has a central role in cellular protein quality control as the main proteolytic system for the ATP-dependent degradation of misfolded proteins and regulated protein turnover. The proteasome consists of the proteolytic core particle (20S) and the regulatory particle (19S), the latter being composed of two subcomplexes, the so-called lid and the ATPase base. In the present study the authors investigated the conformational interplay between lid and base and identified interactions that are critical for engagement and processing of proteasome substrates. Their findings provide new insight into the intricate workings of one of the most complicated macromolecular machines.

**Decision letter after peer review:**

Thank you for submitting your article "Specific lid-base contacts in the 26S proteasome control the conformational switching required for substrate degradation" for consideration by *eLife*. Your article has been reviewed by three peer reviewers, and the evaluation has been overseen by a Reviewing Editor and Cynthia Wolberger as the Senior Editor. The following individuals involved in review of your submission have agreed to reveal their identity: Robert T Sauer (Reviewer #2).

The reviewers have discussed the reviews with one another and the Reviewing Editor has drafted this decision to help you prepare a revised submission.

The reviewers are overall positive and we are happy to invite you to submit a thoroughly revised version of your manuscript. However, the reviewers found various issues with the paper. The most serious of these is that some of the proteasome mutants analyzed have markedly reduced assembly efficiencies, which could explain at least some of the functional effects. In their discussion, all reviewers agreed that final acceptance of the manuscript will hinge on this issue being satisfactorily addressed. One reviewer wrote: "The main concern has to do with Figure 1—figure supplement 2A. This experiment tracks reconstitution of the mutant proteasomes against that of reconstituted wild-type. The Rpn5 VTENKIF and Rpt4 EQ mutants both have quite reduced levels of fully or properly assembled proteasomes. In the case of Rpt4, this reduction is at the expense of a fast-migrating species that is assigned as RP, which is clearly more abundant than the fully assembled proteasome. The three native gels are all consistent so the differences between WT and mutant do not appear to be flukes. Although it is discounted in the main text in several places, some of the behavior of the mutants in various proteasome assays may reflect these differences in assembly, at least in part. For example the Rpt4 mutant is very defective in the degradation of ubiquitinated protein (Figure 3—figure supplement 1), probably some component of this defect reflects of the predominance of free RP. For Rpn5 it is hard for me to tell whether defects such as that shown in Figure 1—figure supplement 1B (25% reduced degradation of ubiquitin-conjugates) result from the altered conformational states or a lower fraction of fully assembled proteasome (looks like more than 25% down). It may or may not be possible in these cases to create mutant proteasomes that are carbon copies of WT but some adjustments or normalizations may make for better data interpretation."

In addition to the assembly issue, we ask that the following major and minor points raised by the reviewers be addressed in the revised manuscript:

1) What should be made of the effect of the Rpt4 mutation on gate opening? The authors dwell on this in the discussion, "the stimulating effects of Rpt4-EQ and Rpt6-EQ mutations on gate opening". From the data in Figure 3D it looks to me that the Rpt4 mutation instead suppresses gate opening (and shows and effect opposite that of Rpt6, not in line with it).

2) In Figure 1A, slow degradation of the substrate by core alone is observed. Isn't this result surprising? Some comment would be useful.

3) Figure 1C shows a large difference between degradation by endogenous and reconstituted proteasomes containing Rpn5-VTENKIF. The authors argue that this is not a consequence of additional Nas6 in the reconstituted prep. Why then is the difference so large?

4) In Figure 1—figure supplements 1A, 1B, and 1D are the proteomes endogenous or reconstituted?

5) In Figure 2, 26S particles were assigned to different classes but in the overlay in panel B, S1 and S3 do not differ dramatically. It would be nice to include: (i) an overlay between previously published high-resolution maps after low-pass filtering with the lower resolution map of the same state; and (ii) an overlay between two lower resolution maps assigned to the same state from different samples to strengthen the argument that the assignment of conformational states are valid at lower resolution and consistent with high-resolution data.

6) In subsection “The nucleotide states of Rpt6 and Rpt4 affect proteasome conformational switching”, it is stated that Rpt3, Rpt6, and Rpt4 contact the TPR domains of the lid. This seems at odds with the cartoon in Figure 3A.

7) In Figure 3C, it would be helpful to show a Ubp6-only control. It is argued that EQ mutations in Rpt6 and Rpt4 bias the conformational distribution and thus increase Ubp6 activity. However EQ mutations in Rpt2, Rpt3, and Rpt5 also increase Ubp6 activity, sometimes as much as EQ-Rpt6 and EQ-Rpt4.

8) In subsection “Proteasomes with biased conformational landscapes display various degradation defects” paragraph two, Figure 4B is referenced with respect to Rpn5-VTENKIF proteasomes but does not contain any information about Rpn5-VTENKIF activity.

9) In Figure 5, why is data for EQ-Rpt6 not shown? Also the data seems to show slow insertion in the presence of ATPgS, but the text (subsection “Disrupting the proteasome conformational equilibrium affects degradation initiation”) states that no tail insertion was observed.

10) It would be useful for readers not familiar with the proteasome literature to have a brief discussion of the differences/similarities between the yeast and human enzymes.

11) Since the negative-stain EM analysis is the major evidence for the Rpn5-VTENKIF mutant perturbing the natural distribution of states, the presentation of the differing states and their clustering shown in Figure 2, Figure 2—figure supplement 1 and Supplementary Figure 3, could be improved for non-experts of proteasome EM. It is stated "Designation of substrate-free and engaged-like conformations was based on a best fit to the atomic models provided in Eisele et al., 2018". Perhaps this best fit can be more emphatically shown or even quantified in the supplementary figures. To the non-expert, it is currently very difficult to discern differences in the different states, even if RP rotation is indicated by the arrow in Figure 2B. This is also true for the comparison of states in Figure 2—figure supplement 1C (grey vs. yellow vs. purple) and the gallery of discarded vs. included classes in Supplementary Figure 3. The alignments were made on the 20S core particle and the authors mention that the relative orientation of the horseshoe-shaped feature of the PCI domains aided in classification – perhaps this can be presented in a more obvious way by coloring or outlining (one clear exemplary figure that demonstrates this alignment can be sufficient as a guide for the others). Also, the authors should comment on how classes were chosen for discarding (red cross on individual classes shown in the gallery in Supplementary Figure 3) – to the non-expert some of these discarded classes look more or less similar to others that were kept.

12) While the Rpn5-VTENKIF mutant and its effect on the lid-base contact network is very clearly illustrated and should be intuitive even for non-experts (aided by models like the one shown in Figure 1B), this is less so for the Rpt6/Rpt4 mutants, owing perhaps to the structural complexity of the AAA+ cycle to non-experts. The authors show in Figure 3A a scheme of base subunit arrangement and contact points with the lid; but perhaps there is a way to utilize existing structures of the AAA+ hexameric staircase and the surrounding lid subunits, to more directly visualize how these two subunits in particular could be pivotal in allowing the base motor to drive the sort of overall conformational switching the manuscript goes to great lengths to analyse. Some passages in the Discussion address the relationship between AAA+ staircase structure and RP conformational states, but the authors could think about including an extra Supplementary Figure to visualize these interaction networks using existing data, even if some points remain speculative.

[Editors' note: further revisions were requested prior to acceptance, as described below.]

Thank you for resubmitting your work entitled "Specific lid-base contacts in the 26S proteasome control the conformational switching required for substrate degradation" for further consideration by *eLife*. Your revised article has been evaluated by Cynthia Wolberger (Senior Editor) and a Reviewing Editor.

The revised manuscript has addressed most of the points raised by the reviewers. The following remaining issues, however, need to be addressed before the paper can be accepted:

1) Figure 1—figure supplement, panels A and B, where B is the quantification of A: B shows about a two-fold effect on the rate of ubiquitin-independent degradation. But the reaction curve appears to be biphasic for VTENKIF, with a visible slow-down in the second phase, beginning at about 600 sec. If you consider the first phase, the initial phase, the defect seems much less. Was the line-fitting done assuming a monophasic decay? Authors please comment.

2) In Figure 1's supplement, panel D, the degradation rate of the reconstituted VTENKIF complex is measured. The significance of this result is described in the main text as having to do with the Nas6 problem (which arises in part from the Tomko paper published earlier this year, where an Rpn5 mutation comparable to VTENKIF was found to impair Nas6 release): these "purified endogenous proteasomes from *S. cerevisiae*" have very little Nas6 and they are nonetheless still degradation-deficient. However, there is no wild-type control (no control at all), so the extent of deficiency cannot be assessed. Authors please add controls for the panels Figure 1—figure supplement 1D and F.

3) There is no legend to Figure 2.

---

## [Author Response]

The reviewers are overall positive and we are happy to invite you to submit a thoroughly revised version of your manuscript. However, the reviewers found various issues with the paper. The most serious of these is that some of the proteasome mutants analyzed have markedly reduced assembly efficiencies, which could explain at least some of the functional effects. In their discussion, all reviewers agreed that final acceptance of the manuscript will hinge on this issue being satisfactorily addressed. One reviewer wrote: "The main concern has to do with Figure 1—figure supplement 2C. This experiment tracks reconstitution of the mutant proteasomes against that of reconstituted wild-type. […] It may or may not be possible in these cases to create mutant proteasomes that are carbon copies of WT but some adjustments or normalizations may make for better data interpretation."

We agree with the reviewers that differential assembly abilities of mutant proteasomes could in theory account for the degradation differences observed in our reconstituted system, and that the native gel presented in the original submission suggested this. However, even though lower or more dynamic RP-CP association may account for some of the defects observed for the Rpt4-EQ variant, we think this is not the case for the variant containing Rpn5-VTENKIF. To strengthen this point, we repeated native-PAGE analyses for all proteasome variants, such that we now have triplicate trials of this experiment, presented in Figure 1—figure supplement 2. It should also be noted that the native-PAGE gels described by reviewers as consistent were in fact images of the same, single gel, visualized by different methods (LLVY hydrolysis -/+ SDS, and Coomassie staining). We have revised the figure legend and text now accordingly to clarify these different readouts for individual gels.

From the additional gels that we produced for this revision, it is clear that Rpn5-VTENKIF-mutant lid can assemble very similarly to wild type (Figure 1—figure supplement 2C). This is further supported by the native-PAGE comparison of wild-type and Rpn5-mutant proteasomes purified as holoenzymes from yeast (new Figure 1—figure supplement 2D), and by the pulldown of wild-type and Rpn5-VTENKIF mutant RP with FLAG-tagged CP (Figure 1—figure supplement 1 F), which represents an alternative non-equilibrium method assessing holoenzyme stability.

The lid titration experiment shown in Figure 1—figure supplement 1C is an equilibrium binding measurement that also indicates similar affinities for wild-type and Rpn5-VTENKIF lids. Mutations in Rpn5 thus do not appear to cause assembly defects that are significant enough to account for the detected degradation defects.

As noted by the reviewers, we do from time to time observe slightly lower amounts of assembled holoenzymes for Rpn5-VTENKIF-containing, reconstituted holoenzyme, which may be a consequence of lingering Nas6. To account for potential small differences in the concentrations of reconstituted holoenzyme (for instance due to the retention of some Nas6), we performed most of the degradation experiment in this manuscript under single-turnover conditions, with excess enzyme relative to substrate, and using concentrations of proteasome subcomplexes that are well above saturating for holoenzyme formation. By doubling the concentration of reconstituted wild-type and Rpn5-VTENKIF proteasomes (new Figure 1—figure supplement 1D), we confirmed that degradation indeed proceeded at maximal velocity and there was no further proteasome assembly at higher concentrations of its subcomplexes. Also, as requested by the reviewers, we confirmed that the addition of excess 19S RP does not inhibit degradation, for instance by sequestering substrates away from assembled holoenzyme (new Figure 4—figure supplement 2F).

As shown in Figure 1C, there is a significant difference in activity between undissociated holoenzymes and reconstituted complexes, whose origin is still unknown. Importantly, however, this difference is observed for wild-type and mutant proteasomes. We thus conclude that the degradation defects of RPN5-VTENKIF-containing proteasomes are caused principally by intrinsic changes in its behavior, e.g. differences in the position and dynamics of conformational equilibria, rather than assembly defects (see also the response to reviewer comment 3).

To address whether assembly defects contribute to the hypo-stimulated gate-opening activity observed for Rpt4-EQ-mutant proteasomes, we compared gate-opening measurements with wild-type, Rpt4-EQ-, and Rpt6-EQ-mutant proteasomes after reconstitution at two different base concentrations and found no change in activities (Figure 3—figure supplement 1). These data suggest that under equilibrium conditions, Rpt4-EQ-mutant proteasomes are likely fully assembled. To address whether the faster migrating species (likely regulatory particle) observed on native gels in Figure 1—figure supplement 2 plays a role in rendering the Rpt4-EQ mutant degradation incompetent, we carried out a series of experiments assessing the effects of additional free regulatory particle on the degradation process of all proteasome mutants (Figure 4—figure supplement 2). We found that compared to the other tested RPs, Rpt4-EQ-containing RP displayed lower Rpn11-mediated deubiquitination activity, which was enhanced by binding to the core particle. The rate of general substrate processing (deubiquitination and complete or partial degradation) by Rpt4-EQ-containing proteasomes was lower than that of nonspecific proteolysis by free CP, indicating that the Rpt4-EQ-mutant RP caps the core particle, hinders access of the substrate’s unstructured tail to the internal degradation chamber, and thus prevents clipping.

These data suggest that Rpt4-EQ-mutant base can assemble into holoenzyme, but lacks engagement and/or translocation properties, rendering the corresponding proteasomes degradation incompetent. We also found that excess RP did not affect the degradation kinetics of other proteasomes tested (wild-type, Rpn5-VTENKIF, and Rpt6-EQ), further supporting that free RP is not responsible for any degradation defects we observed.

In addition to the assembly issue, we ask that the following major and minor points raised by the reviewers be addressed in the revised manuscript:1) What should be made of the effect of the Rpt4 mutation on gate opening? The authors dwell on this in the discussion, "the stimulating effects of Rpt4-EQ and Rpt6-EQ mutations on gate opening". From the data in Figure 3D it looks to me that the Rpt4 mutation instead suppresses gate opening (and shows and effect opposite that of Rpt6, not in line with it).

We thank the reviewers for highlighting this apparent discrepancy. We agree that Rpt4-EQ is insensitive in its gate-opening activity to the addition of ATPγS, and we have revised the text to clarify this.

2) In Figure 1A, slow degradation of the substrate by core alone is observed. Isn't this result surprising? Some comment would be useful.

Ourselves (e.g. Bard et al., 2019) and others (e.g. Myers et al., 2018; Wenzel and Baumeister, 1995) have noted nonspecific proteolysis by the core particle in the absence of regulatory complexes, both in vivo and in vitro, which likely originates from flexible regions of the substrate accessing the proteolytic chamber. To clarify this, we have included a short discussion of this effect in the revised manuscript.

3) Figure 1C shows a large difference between degradation by endogenous and reconstituted proteasomes containing Rpn5-VTENKIF. The authors argue that this is not a consequence of additional Nas6 in the reconstituted prep. Why then is the difference so large?

Based on the comparison of Nas6-free proteasome holoenzymes purified from yeast, it is clear that the significant difference in degradation activities for wild-type and Rpn5-VTENKIF-containing proteasomes indeed originates from intrinsic features, i.e. a biased conformational landscape, of the mutant.

The defects observed for all reconstituted proteasomes compared to non-dissociated holoenzymes may at least in part originate from the retention of Nas6. Proteasomes reconstituted with Rpn5-VTENKIF-mutant lid retain Nas6 to a larger extent and thus show an additional degradation defect relative to their reconstituted wild-type counterparts. The text has been rewritten to clarify this distinction.

There may be additional factors, for instance posttranslational modifications or yet unidentified peptides or subunits, playing a role for the higher activity of endogenous proteasomes, which is an interesting avenue for future exploration, but beyond the scope of this work.

4) In Figure 1—figure supplements 1A, 1B, and 1D are the proteomes endogenous or reconstituted?

We thank the reviewers for pointing out this omission of important experimental details. These experiments were performed with reconstituted proteasomes, and the figure legends have been corrected.

5) In Figure 2, 26S particles were assigned to different classes but in the overlay in panel B, S1 and S3 do not differ dramatically. It would be nice to include: (i) an overlay between previously published high-resolution maps after low-pass filtering with the lower resolution map of the same state; and (ii) an overlay between two lower resolution maps assigned to the same state from different samples to strengthen the argument that the assignment of conformational states are valid at lower resolution and consistent with high-resolution data.

We agree with reviewers that a better representation, showing how proteasome conformations appear at low resolution and comparing these representations to the experimental data, would improve the manuscript. We have therefore updated Figure 2B to better emphasize the conformational changes and replaced Supplemental Figure 3 with Figure 2—figure supplements 2-6. We would like to highlight Figure 2—figure supplement 3 specifically for showing a comparison of simulated 20Å density for the different proteasome conformations compared to each other. Additionally, Figure 2B and Figure 2—figure supplements 4-6 present comparisons between these simulated density maps and experimental density maps.

6) In subsection “The nucleotide states of Rpt6 and Rpt4 affect proteasome conformational switching”, it is stated that Rpt3, Rpt6, and Rpt4 contact the TPR domains of the lid. This seems at odds with the cartoon in Figure 3A.

We thank the reviewers for pointing this out. We have updated the cartoon in Figure 3A to be more accurate and precise regarding the various contacts made between the lid and base, and how these interactions change in different conformations.

7) In Figure 3C, it would be helpful to show a Ubp6-only control. It is argued that EQ mutations in Rpt6 and Rpt4 bias the conformational distribution and thus increase Ubp6 activity. However EQ mutations in Rpt2, Rpt3, and Rpt5 also increase Ubp6 activity, sometimes as much as EQ-Rpt6 and EQ-Rpt4.

As suggested by the reviewers, we have included a Ubp6-only control in Figure 3C. As the Ub-AMC cleavage activity of Ubp6 is stimulated 300-fold by binding to the base (Borodovsky et al., 2001, Leggett et al., 2002, and Bashore et al., 2015), it is not surprising that the activity of Ubp6 in isolation is extremely low.

The Ubp6-stimulation experiments shown in Figure 3 were performed with base in excess over Ubp6 and with a mastermix containing all components except for the base, to ensure that observed differences can indeed be attributed to differential Ubp6 responses to individual base variants rather than differential Ubp6 binding to proteasomes.

We agree with the reviewers that the Ub-AMC activity is elevated for other base variants as well, not just for the two mutants we have chosen to focus on (Rpt4-EQ and Rpt6-EQ). This is in agreement with a previous study (Eisele et al., 2018), indicating that Rpt2-EQ, Rpt3-EQ, and Rpt6-EQ could have aberrant conformational landscapes based on their elevated core gate-opening activity in ATP, crosslinking between an engineered s1-state-specific pair of cysteines, and the appearance of substrate-engaged proteasome conformations in ATP, as observed by cryo-EM. In this study by Eisele et al., only a subset of the Rpt-EQ mutants were tested, and different assays did not always show the same extent of defect. For example, Rpt2-EQ-mutant proteasome exhibited elevated core-gate opening, but no loss of s1-specific crosslinking. We therefore thought it was important to show all single Rpt-EQ mutant proteasomes, but focus further characterization on the mutants Rpt4-EQ and Rpt6-EQ, which displayed consistent defects in both, the Ubp6-stimulated ATPase and Ubp6 Ub-AMC cleavage assays. The effects of ubiquitin-bound Ubp6 on ATPase rates appeared to be the more stringent criterium for the potential of an Rpt-EQ mutation to change the overall conformation of the proteasome.

We have updated the text to include statements about the elevated Ubp6 Ub-AMC activity for other Rpt-EQ mutants and to emphasize that only Rpt6-EQ and Rpt4-EQ were characterized in more detail due to their better agreement between ATPase stimulation and Ubp6 activation.

8) In subsection “Proteasomes with biased conformational landscapes display various degradation defects” paragraph two, Figure 4B is referenced with respect to Rpn5-VTENKIF proteasomes but does not contain any information about Rpn5-VTENKIF activity.

We apologize for this mistake and corrected the error by changing the reference to Figure 4C.

9) In Figure 5, why is data for EQ-Rpt6 not shown? Also the data seems to show slow insertion in the presence of ATPgS, but the text (subsection “Disrupting the proteasome conformational equilibrium affects degradation initiation”) states that no tail insertion was observed.

We have included a comparative data set for substrate-tail insertion into wild-type, Rpn5-VTENKIF, and Rpt6-EQ-mutant proteasomes in Figure 5—figure supplement 1B, showing similar behavior for the two mutant proteasomes.

The reviewers are correct that there is a small extent of tail insertion for the proteasomes in the presence of ATPγS, which is consistent with previous measurements by our lab (Bard et al., 2019) as well as the observations for conformational distributions of proteasomes in ATPγS (Dong et al., 2019). The fraction of proteasomes in the engagement-competent s1 state may in part depend on the extent of ATP contaminations present in ATPγS, but it is generally very low and negligible. We changed the text to more accurately reflect this position.

10) It would be useful for readers not familiar with the proteasome literature to have a brief discussion of the differences/similarities between the yeast and human enzymes.

We agree with the reviewers and have added a brief discussion of the major similarities between yeast and human proteasomes in the Introduction and Discussion section.

11) Since the negative-stain EM analysis is the major evidence for the Rpn5-VTENKIF mutant perturbing the natural distribution of states, the presentation of the differing states and their clustering shown in Figure 2, Figure 2—figure supplement 1 and Supplementary Figure 3, could be improved for non-experts of proteasome EM. It is stated "Designation of substrate-free and engaged-like conformations was based on a best fit to the atomic models provided in Eisele et al., 2018". Perhaps this best fit can be more emphatically shown or even quantified in the supplementary figures. To the non-expert, it is currently very difficult to discern differences in the different states, even if RP rotation is indicated by the arrow in Figure 2B. This is also true for the comparison of states in Figure 2—figure supplement 1C (grey vs. yellow vs. purple) and the gallery of discarded vs. included classes in Supplementary Figure 3. The alignments were made on the 20S core particle and the authors mention that the relative orientation of the horseshoe-shaped feature of the PCI domains aided in classification – perhaps this can be presented in a more obvious way by coloring or outlining (one clear exemplary figure that demonstrates this alignment can be sufficient as a guide for the others). Also, the authors should comment on how classes were chosen for discarding (red cross on individual classes shown in the gallery in Supplementary Figure 3) – to the non-expert some of these discarded classes look more or less similar to others that were kept.

We thank the reviewers for criticizing this shortcoming and agree that it is useful to show additional details regarding the methodology of picking conformations. We have updated Figure 2B to now depict a comparison of low-resolution density maps from previous studies overlaid with our experimental maps. In addition, Supplementary Figure 3 was replaced with Figure 2—figure supplements 2-6, which address all of the points suggested by reviewers. Together with our response to point number 5 above, we hope that these new Supplementary figures make it clear how the classes were derived and what structural features distinguish different conformations.

We also want to point out that there was a mistake in the originally submitted Supplementary Figure 3, reporting incorrect numbers of discarded classes. This had no effect on the results or interpretations, but has now been corrected in the new supplementary figure.

12) While the Rpn5-VTENKIF mutant and its effect on the lid-base contact network is very clearly illustrated and should be intuitive even for non-experts (aided by models like the one shown in Figure 1B), this is less so for the Rpt6/Rpt4 mutants, owing perhaps to the structural complexity of the AAA+ cycle to non-experts. The authors show in Figure 3A a scheme of base subunit arrangement and contact points with the lid; but perhaps there is a way to utilize existing structures of the AAA+ hexameric staircase and the surrounding lid subunits, to more directly visualize how these two subunits in particular could be pivotal in allowing the base motor to drive the sort of overall conformational switching the manuscript goes to great lengths to analyse. Some passages in the Discussion address the relationship between AAA+ staircase structure and RP conformational states, but the authors could think about including an extra Supplementary Figure to visualize these interaction networks using existing data, even if some points remain speculative.

We thank the reviewers for pointing out this need for better representations of the interaction networks within the proteasome’s regulatory particle. We have updated Figure 3A to indicate which ATPase pockets are in ‘open’ versus ‘closed’ configurations for “substrate-free” (s1) and “substrate-engaged” (e.g. s3) conformations. This representation emphasizes the open-pocket conformation for Rpt6 in substrate-free states and visually aids our discussions of this ATPase pocket contributing to changes in the proteasome’s conformational landscape. For structural models of the Rpt6 ATPase pocket and how it relates to proteasome conformation, we would like to refer readers to the cited literature.

[Editors' note: further revisions were requested prior to acceptance, as described below.]

The revised manuscript has addressed most of the points raised by the reviewers. The following remaining issues, however, need to be addressed before the paper can be accepted:1) Figure 1—figure supplement, panels A and B, where B is the quantification of A: B shows about a two-fold effect on the rate of ubiquitin-independent degradation. But the reaction curve appears to be biphasic for VTENKIF, with a visible slow-down in the second phase, beginning at about 600 sec. If you consider the first phase, the initial phase, the defect seems much less. Was the line-fitting done assuming a monophasic decay? Authors please comment.

The reviewer is correct that the initial slope for the VTENKIF-mutant trace does not reflect a 40% lower rate compared to wild type, as shown in panel B. The rates in panel B originate from linear regressions between 15 s and 570 s for traces of three replicates, including the one shown in panel A. Unfortunately, that depicted trace has the highest slope (~ 80% of wild type) and in addition a kink at ~ 600 s. To better represent our data and the lower activity of the VTENKIF mutant, we now picked a different example trace that is linear over the entire measurement and indicates a ~ 50% lower rate compared to WT. Shown in the graph in Author response image 1 is a comparison of this new trace with the trace previously shown in panel A. Both traces, as well as a third repetition, were considered for the determination of the degradation rates and standard errors shown in panel B.

2) In Figure 1's supplement, panel D, the degradation rate of the reconstituted VTENKIF complex is measured. The significance of this result is described in the main text as having to do with the Nas6 problem (which arises in part from the Tomko paper published earlier this year, where an Rpn5 mutation comparable to VTENKIF was found to impair Nas6 release): these "purified endogenous proteasomes from S. cerevisiae" have very little Nas6 and they are nonetheless still degradation-deficient. However, there is no wild-type control (no control at all), so the extent of deficiency cannot be assessed. Authors please add controls for the panels Figure 1—figure supplement 1D and F.

We thank the reviewer for pointing this out. The confusion about missing controls likely originated from a mistake in the figure reference, where we referred to Figure 1—figure supplement 1D instead of the correct Figure 1C to show consistent degradation defects for endogenous VTENKIF proteasome holoenzymes. This mistake has been corrected in the revised manuscript. In addition, we have now included in Figure 1—figure supplement 1D (now panel C) the traces for single-turnover substrate degradation by reconstituted wild-type proteasomes, measured under the same conditions as the Rpn5-VTENKIF mutant. For both variants, doubling the concentration of holoenzyme (2X) did not change the observed kinetics, confirming that even at ‘1X’ enzyme the degradation occurred in single turnover.

Otherwise, we are not aware of any wild-type controls missing from Figure 1C or the Figure 1—figure supplement 1 panels E and F. Both, panel E and F, show a direct comparison of the Nas6 retention for wild-type and VTENKIF-mutant proteasomes during in vitro reconstitution as well as for purified endogenous holoenzymes.

3) There is no legend to Figure 2.

We apologize for this mistake and have now included the legend to Figure 2.